



# Uncertainty sources in a large ensemble of hydrological projections: Regional Climate Models and Internal Variability matter

Evin Guillaume[1], Hingray Benoit[1], Thirel Guillaume[2,3], Ducharne Agnès[4], Strohmenger Laurent[2], Corre Lola[5], Tramblay Yves[6], Vidal Jean-Philippe[7], Bonneau Jérémie[7,8], Colleoni François[9], Gailhard Joël[10], Habets Florence[11], Hendrickx Frédéric[12], Héraut Louis[7], Huang Peng[4], Le Lay Matthieu[10], Magand Claire[13], Marson Paola[14], Monteil Céline[12], Munier Simon[5], Reverdy Alix[1], Soubeyroux Jean-Michel[14], Robin Yoann[15], Vergnes Jean-Pierre[16], Vrac Mathieu[15], and Sauquet Eric[7]

[1]Univ. Grenoble Alpes, INRAE, CNRS, IRD, Grenoble INP, IGE, Grenoble, France
[2]Université Paris-Saclay, INRAE, UR HYCAR, Antony, France
[3]Univ Toulouse, CNES/IRD/CNRS/INRAE, CESBIO, Toulouse, France
[4]Sorbonne Université/CNRS/EPHE, METIS-IPSL, Paris, France
[5]CNRM, Météo-France, CNRS, Université de Toulouse, Toulouse, France
[6]UMR Espace Dev (Univ. Montpellier, IRD), Montpellier, France
[7]UR RiverLy, INRAE, Villeurbanne, France
[8]INSA Lyon, DEEP, UR 7429, Villeurbanne, France
[9]UMR RECOVER, INRAE, Aix-Marseille University, Le Tholonet, France
[10]Département Eau Environnement, EDF-DTG, Saint Martin le Vinoux, France
[11]Geology Laboratory of Ecole Normale Supérieure, Pierre Simon Laplace Research University, CNRS UMR 8538, Paris, France
[12]Département LNHE, EDF-R&D, 78401 Chatou, France
[13]OFB, Direction de la recherche et de l'appui scientifique, Nantes, France
[14]Météo-France, Direction de la Climatologie et des Services Climatiques, Toulouse, France
[15]LSCE, IPSL, CEA-CNRS-UVSQ, Univ. Paris-Saclay, Gif-sur-Yvette, France
[16]BRGM – French Geological Survey, Orléans, France

**Correspondence:** Evin Guillaume (guillaume.evin@inrae.fr)

**Abstract.** Multi-scenario, multi-model ensembles of hydrological projections are widely used to describe possible futures of regional hydrology and inform adaptation strategies. The Explore2 dataset is such an ensemble of river flow projections in Metropolitan France. It provides future simulations for 1,735 catchments with modeling chains composed of different hydrological models forced by 36 regional climate projections based on bias-adjusted EUROCORDEX simulations. This study
5    assesses the uncertainties of this ensemble with QUALYPSO, a method specifically designed to deal with incomplete ensembles and to disentangle and quantify all uncertainty sources, including that due to internal variability.

Focusing on results obtained at the end of the century, this study shows a strong agreement between modeling chains towards decreases in low flows in a large southern part of France for a high-emission scenario, and very uncertain changes for the annual mean and high flows. Emission scenario uncertainty is the dominant source of uncertainty for low flows over the
10    whole of France, and for mean annual flows in southeastern France. The contribution of the global and regional climate models is important for mean and high flows, especially in rainfall-dominated areas. Regional climate models contribute considerable uncertainty to low flows, much more than global models. The contribution of hydrological model uncertainty is large for low





flows, moderate for mean annual flows, and small for high flows. For all climate and hydrological indicators, internal variability is often large and cannot be overlooked. It is often of the same order and sometimes larger than the uncertainty on the climate change response.

## 1 Introduction

Hydrological projections for future climates are often based on modeling chains driven by emission scenarios depicting future emissions of greenhouse gases and aerosols. Modeling chains are composed of a global climate model (GCM) to simulate global climate evolution for the chosen scenario, a regional climate model (RCM) to refine the simulation for a specific region, a bias adjustment model (BAM) to adjust systematic errors in the regional climate scenario, and a hydrological model (HM) to assess hydrological impacts of projected climate changes for the targeted river basin. Modeling chains made of different scenarios or different models for each category of models are expected to project different climate responses to emission scenarios, i.e. different long-term evolutions of climate or hydroclimatic variables. Multi-scenario, multi-model ensembles of projections (MMEs) provide the opportunity to characterize the spread and the degree of agreement between projections as well as the different sources of uncertainty in the projections (Hawkins and Sutton, 2009; Lehner et al., 2020; Evin et al., 2021).

Sources of uncertainty include uncertainty on future emissions (i.e. scenario uncertainty), model uncertainty, and internal variability of the climate system (see, e.g., Hawkins and Sutton, 2009). Future emissions of greenhouse gases and aerosols for the coming decades are highly uncertain. They will depend on a range of different socio-economic factors and decisions around the world. In the Coupled Model Intercomparison Projects (CMIPs) designed to improve knowledge on climate change and its impacts, climate projections are produced for different emission scenarios (e.g. RCP and SSP scenarios, Moss et al., 2010; Riahi et al., 2017).

Model uncertainty arises from model imperfections. Different models of the same natural system (e.g. climate models) simulate different responses to the same forcing scenario. As shown by COordinated Regional Downscaling EXperiments (CORDEX), where regional projections are obtained from multiple combinations of GCMs and RCMs, both GCMs and RCMs can have important contributions to model uncertainty (Bichet et al., 2020; Evin et al., 2021; Christensen and Kjellström, 2021). In MMEs obtained with multiple HMs, a substantial contribution can also come from HM uncertainty (Lafaysse et al., 2014; Chegwidden et al., 2019; Lemaitre-Basset et al., 2021; Aitken et al., 2023). Scenario uncertainty and model uncertainty make the climate response uncertain for all climate or hydroclimatic variables.

Additional uncertainty arises from climate internal variability (IV), resulting from the chaotic and nonlinear nature of the climate system (Deser et al., 2012). IV introduces erratic multiscale fluctuations to the climate response, potentially causing unusual sequences of meteorological events, unusual years, or sequences of years. Unlike scenario or model uncertainty, IV is irreducible. It decreases with temporal and/or spatial aggregation. IV is small for temperature-related indicators but significant for precipitation and hydrology-related indicators (Hawkins and Sutton, 2009; Lehner et al., 2020; Evin et al., 2021; Vicente-Serrano et al., 2025). While it can be an important component of the overall uncertainty in hydrological projections (Lafaysse et al., 2014; Vidal et al., 2016; Chegwidden et al., 2019; Alder and Hostetler, 2019), IV is often overlooked in impact studies.





However, accounting for IV is crucial for designing robust adaptation strategies, particularly in addressing extreme or atypical conditions.

The estimation of scenario uncertainty, model uncertainty components, and internal variability is often performed by applying Analysis of Variance (ANOVA) methods to large MMEs that combine multiple models across different emission scenarios.
However, this estimation faces two challenges:

- **Filtering out IV fluctuations for climate response estimation:** Disentangling the climate response of a given chain from stochastic fluctuations caused by IV is key for a relevant uncertainty analysis. Estimating the climate response can be challenging, particularly for indicators such as precipitation, where IV is significant (Hingray et al., 2019). This difficulty arises because climate outputs blend the climate responses with chaotic fluctuations from IV, which propagate
through all the subsequent models in the chain. If for a given GCM multiple members are available and used for subsequent simulations, the climate response of a modeling chain forced by this GCM can be estimated with the multi-member mean of the simulations, and IV can be estimated with the inter-member variability. However, many hydrological studies rely on single-member and time-slice GCM experiments. As a consequence, IV cannot be properly filtered out and, when they are not simply disregarded, stochastic fluctuations from IV are often attributed to GCM uncertainty (see, e.g.,
Bosshard et al., 2013; Vetter et al., 2017; Gangrade et al., 2020).

- **Dealing with incomplete or unbalanced MMEs:** MMEs are often incomplete or unbalanced. This is usually the case when climate projections are simulated by RCMs since many GCM/RCM combinations are often missing in GCM/RCM MMEs. At the same time, some models of a given model category (GCM, RCM) are often over- or under-represented in MMEs (see, e.g., the large EUROCORDEX MME assessed in Evin et al., 2021). When MMEs are incomplete or
unbalanced, uncertainty analysis cannot be performed with a naive ANOVA approach, unless MMEs are subsampled to produce a complete and balanced MME subset, but this can lead to a dramatic waste of information (Tramblay and Somot, 2018; Christensen et al., 2019).

The "Quasi-Ergodic Analysis of Climate Projections Using Data Augmentation" approach (QUALYPSO) was specifically designed to address these two challenges (Evin et al., 2019). Based on the quasi-ergodic assumption for transient climate
projections (Hingray and Saïd, 2014), QUALYPSO separates long-term climate responses from multiscale fluctuations caused by IV and applies an ANOVA to the differences of the climate responses between future and reference periods, a.k.a. climate change responses (CCRs). This allows disentangling all uncertainty sources, including IV. QUALYPSO can also be applied to incomplete and unbalanced MMEs, providing thorough uncertainty analyses even in those configurations, as is often the case for hydroclimatic ensembles derived from CORDEX experiments. QUALYPSO has been used in various contexts, for regional
climate projections in Europe (Evin et al., 2021), Africa (Bichet et al., 2020) and for climate-related indicators in different sectoral applications such as renewable energy (Bichet et al., 2019) or hydrology (Lemaitre-Basset et al., 2021; Aitken et al., 2023; Jeantet et al., 2023; Thirel et al., 2025).

The present study aims to characterize uncertainty components in the Explore2 MME, a very large ensemble of climate and hydrological projections developed for France from EUROCORDEX projections (Marson et al., 2024; Sauquet et al., 2025).





The Explore2 MME was developed as part of the Explore2 research project. This project, funded by the French Ministry of Ecological Transition (MTECT) and the French Biodiversity Agency (OFB), aims to support water management adaptation for French rivers in the 21st century (Sauquet et al., 2025). To our knowledge, the Explore2 MME is the largest ensemble of hydrological projections ever produced from regional climate experiments at the scale of a country. Analyses conducted across numerous catchments throughout an entire country are particularly valuable for exploring how projected changes vary geographically, as they reflect differences in hydroclimatic conditions, sensitivities, and regional climate shifts (Addor et al., 2014; Chegwidden et al., 2019; Aitken et al., 2023). This study leverages the large Explore2 MME and the wide hydroclimatic diversity in France to address the following questions:

- What are the projected future changes for different hydrological metrics for France and how much do the modeling chains agree on the projections?

- How large is the IV compared to CCR uncertainty?

- How do scenario and model uncertainties contribute to the CCR uncertainty?

- What is the influence of individual models on projections (so-called main effects), and are there important discrepancies between them?

- How do results vary by location and hydroclimatic context?

Section 2 summarizes the different hydrological regimes encountered in France, describes the Explore2 MME and provides a summary of the climate and hydrological models used to produce the MME. It also describes QUALYPSO, the method used for the uncertainty analysis. Section 3 presents projected changes and related uncertainties for different climate and hydrological indicators. Section 4 highlights the key features of the different sources of uncertainty. Section 5 further discusses these results and some related aspects. Section 6 concludes.

## 2 Data and methods

### 2.1 Hydrology of French rivers

France presents a large panel of hydrological regimes resulting from various physiographical contexts (e.g. geology, topography, see Fig. 1a) but also from the large diversity of regional climates (oceanic influences in the West, Mediterranean in the South-East, Strohmenger et al., 2024). Catchment regimes can be considered from a hydroclimatic perspective, as provided by the Budyko (1974) framework. The Budyko model, which provides a simplified representation of mean annual water balance, allows analyzing interactions between energy and water limitations and their impact on catchment hydrology. More specifically, the ratio between total precipitation (P) and potential evapotranspiration (PET) provides a humidity index that can be used as a proxy of different types of interactions and different impacts on runoff. P/PET ratio values lower than one characterize water-limited regions, where dry conditions prevail and sensible heat fluxes are typically high. In contrast, P/PET ratios greater





than one denote energy-limited regions, which are generally wetter and exhibit lower sensible heat fluxes (Sposito, 2017). In France, most regions are energy-limited, especially mountainous areas (Alps, Jura, Vosges in the East, Pyrénées in the South, Massif Central in the center, see Fig. 1b). Water-limited regions are found in the South (Mediterranean region), in a large area northwest of the Massif Central and in some small areas. As reported by Gan et al. (2021) or Lemaitre-Basset et al. (2022), the projected evolution of hydrological changes can be interpreted according to this distinction.

Hydrological regimes can also be classified based on the seasonal cycle of inter-annual mean monthly discharges, following the work of Pardé (1933). Sauquet et al. (2024) propose seven regimes ranging from snow-dominated regimes in mountainous areas to mixed regimes downstream and/or rainfall-dominated regimes in lowlands (Figure 1c,d). Snow-dominated regimes are mainly characterized by a monthly peak flow in spring due to snowmelt which can be particularly large for high-elevation mountainous catchments. Three different snow-dominated regimes are considered: 1) a regime with a dominant nival con-

tribution (N) found in the highest regions of the Alps, 2) a Nivo-Pluvial regime (NP) observed at downstream locations, in mid-elevation areas (e.g. Pyrénées, PreAlps regions) and in southeastern France where Mediterranean events can be important, and 3) a Pluvial-Nival (PN) regime mainly found in downstream locations and low-elevation mountainous areas (Massif Central, Vosges, Jura, Corsica).

Rainfall-dominated regimes present a seasonality mostly derived from the seasonality of precipitation and evapotranspiration

losses, with high flows in winter and low flows in summer. The winter / summer contrast depends on the catchment. It is very large in the Western and northeastern parts of France where catchments have a low storage capacity, large in the North and the South-West, moderate in the Center and the East, and very small in regions where river streamflows are sustained all year long by important aquifers (e.g. in the Paris Basin region). This diversity is represented by four regimes: highly contrasted pluvial (HCP), contrasted pluvial (CP), pluvial (P) and non-contrasted pluvial (NCP). Different hydrological responses to climate

change are expected according to these regimes (e.g., Sauquet et al., 2024).





**Figure 1.** (a) Physical Map of Metropolitan France. (b) P/PET ratio values indicating limiting conditions as estimated for the reference period 1976-2005 from SAFRAN meteorological reanalyses. Ratios greater than one (green colors) indicate energy-limited regions and ratios smaller than one (brown colors) indicate water-limited regions. (c) The seven main hydrological regimes found in Metropolitan France, classified according to the annual cycle of monthly discharges. They consist of four rainfall-dominated regimes: highly contrasted pluvial (HCP), contrasted pluvial (CP), pluvial (P) and non-contrasted pluvial (NCP) and three snow-dominated regimes: Nival (N) , Nivo-Pluvial (NP), Pluvial-Nival (PN). (d) Hydrological regimes estimated for the 1735 simulation points considered in this study. Estimation from discharge time series simulated for the reference period 1976-2005 from SAFRAN meteorological reanalyses (Vidal et al., 2010). See Sauquet et al. (2024) for details.





## 2.2 Explore2: a large climate and hydrological multi-model ensemble

The Explore2 dataset, spanning the period 1976–2099, was generated using various scenarios and models shown in Figure 2. A detailed description is provided in Sauquet et al. (2025). The 36 climate projections correspond to a subset of the large CMIP5-EUROCORDEX ensemble (Jacob et al., 2014, 2020) and have been obtained with three emission scenarios (10 with the RCP2.6, 9 with the RCP4.5, and 17 with the RCP8.5, see Marson et al., 2024). It is based on six CMIP5 GCMs (Taylor et al., 2011) downscaled by nine RCMs. As summarized in Table 1, it considers 17 GCM/RCM combinations and is therefore incomplete (e.g. not all GCMs have been used to force a given RCM) but the selected projections ensure a diverse range of climate models and include at least two simulations for each GCM and RCM. This selection enhances the robustness of the uncertainty component estimates.

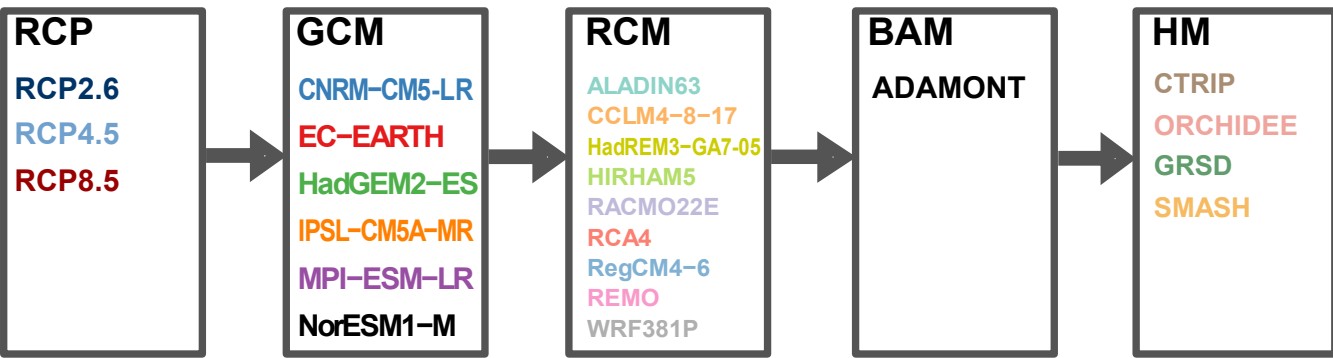

**Figure 2.** Modeling chain: Emission scenarios (RCP), Climate models (GCM, RCM), Bias Adjustment Model (BAM) and Hydrological Models (HM) considered in this study.

The BAM ADAMONT (Verfaillie et al., 2017) was applied to adjust climate projections for some systematic errors using the meteorological reanalysis SAFRAN (Vidal et al., 2010; Le Moigne et al., 2020) as the reference dataset. Adjusted climate projections are produced on a regular horizontal 8 km grid covering Metropolitan France and a part of Switzerland. In this study, we focus on the 8981 grid points that cover Metropolitan France. Reference evapotranspiration (ET0) is obtained from the Penman-Monteith formula (Allen et al., 1998) with a parameterization for short grass. The net radiation is derived from the Hargreaves equation (Hargreaves and Samani, 1985).

Explore2 hydrological projections offer a 'natural reference hydrology' as a foundation for future studies on water allocation. Streamflow simulations are provided in the absence of any water management interventions—such as abstractions, releases, or reservoir operations—thus representing natural flow conditions. In this study, we assess an ensemble of hydrological projections obtained for 1735 simulation points with four HMs of different types (Table 2) covering almost all the country, namely CTRIP, GRSD, ORCHIDEE and SMASH (Fig. 1). The Explore2 MME provides hydrological simulations with nine HMs, but the five other HMs cover only specific regions of France and for fewer catchments (Sauquet et al., 2025). SMASH and GRSD HMs are conceptual "bucket" type models. CTRIP and ORCHIDEE are land surface models and physically based. No single model provided discharge simulations for all simulation points across the study area. Some simulation points could not





| RCM \ GCM | CNRM-CM5-LR | EC-EARTH | HadGEM2-ES | IPSL-CM5A-MR | MPI-ESM-MR | NorESM1-M |
|---|---|---|---|---|---|---|
| ALADIN63 | ⊛ | | ○ | | | |
| CCLM4-8-17 | | | ⊕ | | ⊛ | |
| HadREM3-GA7-05 | ○ | ⊗ | ⊗ | | | |
| HIRHAM5 | | | | ○ | | ⊕ |
| RACMO22E | | ⊛ | | | | |
| RCA4 | | ⊛ | | ⊕ | | |
| RegCM4-6 | | | ⊗ | | ⊗ | |
| REMO | | | | | ⊛ | ⊛ |
| WRF381P | | | | | | ○ |

**Table 1.** Unbalanced ensemble of EUROCORDEX GCM/RCM combinations used for Explore2 climate projections with the scenarios RCP2.6 (×), RCP4.5 (+) and RCP8.5 (○). Only one run was used for each GCM (member r1i1p1 for all GCMs except for EC-EARTH where the member r12i1p1 has been used). All HMs were forced by all climate projections.

155  be reliably paired with their corresponding locations in the hydrographic network (or were excluded due to large differences in simulated versus physical catchment areas). All HMs used the same meteorological forcings: the reanalysis SAFRAN as the reference dataset for the period 1976-2005 and adjusted scenarios for historical and future climate experiments. Calibration of each HM, if required, was model-specific and carried out automatically or manually based on the expertise of the model developers. To assess model performance, all HMs have been evaluated within a standardized framework, focusing on their ability to replicate various observed hydrological signatures for the current climate (see Sauquet et al., 2025, for evaluations).

| Name | # points | Reference |
|---|---|---|
| **CTRIP** | 2024 | Munier and Decharme (2022) |
| **GRSD** | 3712 | de Lavenne et al. (2019) |
| **ORCHIDEE** | 3587 | Huang et al. (2024) |
| **SMASH** | 3821 | Jay-Allemand et al. (2020); Huynh et al. (2024) |

**Table 2.** Names of the surface hydrological models, the associated number of streamflow simulation points and the corresponding reference.

160  We carried out our analyses for a selection of indicators representative of projected changes in climate and hydrology. Indicators considered in the present work are surface air temperature (TAS), total precipitation (PR), reference evapotranspiration (ET0) at a seasonal scale (summer and winter), annual maximum daily precipitation (RX1D), annual minimum monthly flow (QMNA), mean annual daily discharge (QA), and annual maximum daily discharge (QJXA).





## 2.3 Multi-model ensemble characterization with QUALYPSO

To characterize how a given climate or hydrological indicator is projected to change, we use the QUALYPSO method (Evin et al., 2019). QUALYPSO focuses on the climate change response (CCR) projected for the considered indicator. The CCR projected by a given simulation chain defines, for any future time, the absolute or relative difference of the long-term climate response for this future time compared to the reference period. The QUALYPSO method is implemented in an R package (Evin, 2023) and was specifically designed to tackle the main challenges of uncertainty partitioning in unbalanced and incomplete MMEs. More information is provided in Appendix A.

QUALYPSO aims to estimate, for any future time, the CCR of each modeling chain, the mean CCR of the MME, the dispersion between the CCRs of individual chains, the sources of uncertainty explaining this dispersion, and the extra potential dispersion in future realizations of climate indicators which can result from IV.

### 2.3.1 Main assumptions

Following Hawkins and Sutton (2009) and Hingray and Saïd (2014), QUALYPSO is based on two main hypotheses, expressed as follows:

1. The climate response of a simulation chain $i$ has a temporal variation that is inherently gradual and smooth and corresponds to the long-term trend of the simulated projection. The high- to mid-frequency fluctuations in the simulated projections result solely from IV.

2. The CCR of simulation chain $i$ can be expressed as a linear sum of the ensemble mean CCR and of the main effects of the different components of the chain (main effects of the scenario $s$, GCM $g$, RCM $r$ and HM $h$).

The first assumption implies that it is reasonable to consider a trend model to estimate the climate response of a chain, and in turn fluctuations due to IV (deviations from the climate response). Hingray et al. (2019) have shown that this assumption allows providing, for all uncertainty components, more precise estimates than estimates obtained with time-slice approaches.

The additive and linear decomposition model of the second assumption can be derived for any future time using a fixed-effects ANOVA model (see Eq. A6 in the Appendix). In the case of a complete MME, the main effect of a given scenario or model is easily interpreted as the mean difference between 1) the CCR of all the projections using this scenario or model and 2) the mean CCR of the ensemble.

### 2.3.2 Estimation of the climate response

In this study, the climate response of each modelling chain is estimated by applying a trend model to the corresponding projection available for 1976-2099. Cubic smoothing splines are considered (function `smooth.spline` in R {R Core Team}, 2022). For all indicators except temperature, IV is relatively large compared to the long-term trend, so the smoothing parameter `spar` was set to 1.1 to reduce the model's flexibility. This prevents misattributing the low-frequency fluctuations caused by IV



to the climate response. For temperature, we apply a lower smoothing parameter value of 1 to provide more flexibility (Evin
et al., 2021).

### 2.3.3 Estimation of the climate change response and of internal variability

The CCR of each chain is obtained by computing the differences in the climate response between future periods and the
reference period 1976-2005. Absolute changes are considered for temperature (in °C) and relative changes otherwise (in %).
IV contribution is assumed to be constant over time, although this assumption could be relaxed as done in Hingray and Saïd
(2014). For a given chain $i$, we estimate IV as the standard deviation of the annual deviations from its climate response. IV
varies from one chain to another. Here, we use the IV averaged over the different simulation chains to characterize the IV
component of the MME.

### 2.3.4 ANOVA model

A fixed-effect ANOVA model is used to estimate the ensemble mean CCR and the main effects of all scenarios and models
belonging to the different scenario and model categories. The CCR uncertainty of a given category (scenario uncertainty, GCM
uncertainty, RCM uncertainty, HM uncertainty) is estimated by the variance of the main effects of all models (or scenarios)
belonging to this category. The residuals of the ANOVA model (residual variability, see Eq. A6) arise from the imperfect
estimation of the climate responses and limitations of the additive assumption. The residuals thus include potential interactions
between scenarios and models and/or between models of different categories (e.g. Hingray and Saïd, 2014; Evin et al., 2021).

### 2.3.5 Characterization of the uncertainties

In the following, to characterize the CCR uncertainty of the MME, we use the square root of the CCR uncertainty variance,
calculated as the sum of the uncertainty variances of the different CCR uncertainty sources (scenarios, GCMs, RCMs, and
HMs) and of the residual variance of the ANOVA model. The CCR uncertainty is thus expressed in the unit of the considered
indicator.

Results of QUALYPSO analyses will be summarized, for each climate and hydrological indicator, with different metrics:

- The 5 %, 50 %, and 95 % quantiles of the ensemble of CCR. They provide a quick overview of both the MME mean
  CCR and its dispersion.

- The CCR uncertainty and the IV of the MME. The CCR uncertainty also gives a synthetic view of the dispersion
  between projected CCRs. IV indicates the non-predictable variability component due to climate natural variability. IV is
  also presented as standard deviation and expressed in the unit of the considered indicator. CCR uncertainty and IV can
  thus be compared.





– The percentage variance contribution to CCR uncertainty variance (in %) of each CCR uncertainty source (scenario, GCM, RCM, HM) and unexplained CCR uncertainty (residual variability, RV). This indicates the relative importance of the different sources and the dominant ones.

Note that the CCR uncertainty can be compared to the ensemble mean CCR, giving also an indication of the overall spread of the CCR compared to the mean change. If the CCR follows a normal distribution (which is roughly the case for the indicators considered here), 67 % of the simulation chains would have CCR values in the interval [Q50-CCRU, Q50+CCRU] on average, where Q50 is the median CCR and CCRU is the CCR uncertainty. All these metrics vary as a continuous function of future time. In the following, we mainly present results obtained for the end of the century (period 2071-2099).

## 3 Climate change response, associated uncertainty, and internal variability

In this section, we first present results for climate indicators obtained from the MME composed of 36 Explore2 transient climate projections by applying QUALYPSO to each of the 8981 grid points covering France. We next show the results for three hydrological indicators, namely the annual minimum monthly flow (QMNA), the mean annual discharge (QA) and the annual maximum daily discharge (QJXA). These indicators can be considered representative of low flows, mean discharge, and high flows, respectively. QUALYPSO is applied to the MME for each of the 1735 simulation points available for the four HMs. In this study, each MME is thus composed of 144 transient hydrological projections obtained with the four HMs forced by the 36 climate projections. We present the changes projected at the end of the century (2071-2099) with respect to the reference period 1976-2005. For each indicator, we show the mean projected changes and the inter-model dispersion with the scenario RCP8.5 which corresponds to greater changes in climatic and hydrological variables. This section then shows how IV compares to CCR uncertainty and what climate changes are projected for lower emission scenarios RCP2.6 and RCP4.5.

### 3.1 Climate change response projected in France

Figure 3 shows the 5 %, 50 %, and 95 % quantiles of the CCR for the climate indicators, for the RCP8.5. These results confirm those reported in previous publications based on EUROCORDEX ensembles (e.g. Jacob et al., 2014; Christensen and Kjellström, 2020; Coppola et al., 2020; Evin et al., 2021; Marson et al., 2024; Corre et al., 2025). Projected climate change depends on the indicator, season, and region. It is contrasted and uncertain.

Temperature is projected to increase significantly, especially in summer. Depending on the region, the median projected increase varies from 2.5 °C to 4.5 °C in winter and from 3.5 °C to 6 °C in summer. Summer warming is projected to be much greater in mountainous areas (Alps, Pyrénées, Massif Central, Vosges) and in the south, particularly in the South-East. The dispersion between modeling chains is important. The 5 % - 95 % interquantile range is around 1.5 °C in winter, up to 4 °C in summer. The largest warming projected in summer is greater than 7.5 °C.

In summer, precipitation is projected to significantly decrease, from -10 % to -40 % for the median CCR depending on the region. Although there is considerable dispersion among the projections, they largely agree on the direction of change. However, a few exceptions exist, notably in the Rhône River valley and some eastern regions, where some projections indicate





a slight increase. The most extreme projections lead to very large decreases, -60 % on average over France and up to -85 % in
the South.

In contrast to summer, winter precipitation is mostly projected to increase (but less than the projected decrease in summer).
The median change is around 0 % in the South, South-East and exceeds 30 % over the northern half. The inter-model dispersion
remains significant, although it is less pronounced than in summer. Models almost always agree on the sign of the changes and,
except in the South, even the 5 % quantile projects slight increases.

The median reference evapotranspiration (ET0) is projected to increase, from 20 % to 30 % over most of France (and slightly
more in summer). If all models agree on the sign of changes, the inter-model dispersion is large. In summer, the 5 % quantile
ranges from 10 % to 20 % and the 95 % quantile is larger than 40 %. In general, results roughly follow those obtained for
temperature, but spatial patterns of changes do not match the N-W, S-E gradients obtained for temperature. The South-East
and the mountainous areas do not appear as hot spots of changes.

The median annual maximum daily precipitation is projected to increase. Projected increases vary a lot from one location
to the other, from 0 % to 20 % in a large southern part to (20 %, 30 %) in northern areas. Whatever the region, the inter-model
dispersion is very large. In the south, for instance, models do not agree on the sign of the changes, and the most extreme chains
project decreases down to (-40 %, -20 %) and large increases up to (40 %, 60 %).





**Figure 3.** 5 %, 50 %, and 95 % quantiles of the climate change responses for seasonal temperature (TAS), precipitation (PR), reference evapotranspiration (ET0) and annual daily precipitation maxima (RX1D) changes (2071-2099 relative to 1976-2005) in summer (JJA) and winter (DJF) for the RCP8.5.



**Figure 3.** (continued).





### 3.2 Hydrological response to climate change projected in France

The 5 %, 50 %, and 95 % quantiles of the CCR for the hydrological indicators, for the RCP8.5 (Figure 4) are contrasted, uncertain, and depend on the region.

Low flows are mostly projected to decrease, from -10 % in the North to (-60 %, -45 %) in the South for the median CCR. While the inter-model dispersion is very large, projections agree on a decrease for most southern regions (with the exception of the Alps). The driest chains lead to very large decreases, greater than -60 % (except for a small area in the North where they 275 "only" reach -45 % to -60 %). Conversely, the most humid ones project large increases (40 % or more) in the Center-North.

For median annual discharges, while slight increases are projected (often less than 15 %) in the North, decreases (up to -30 %) are projected in the South. With the exception of the Pyrénées and the southern Alps, where most models project a decrease, modeling chains do not agree on the sign of change. The inter-model dispersion is very important. Large decreases are projected for the driest chains (up to -75 % in the South-East) while large to very large increases are projected for the most 280 humid ones (up to 75 % in the North).

Results for high flows are roughly similar. The median high flow is projected to slightly increase (up to 15-30 %) in the North and to decrease in the South. Whatever the region, the inter-model dispersion is large, and models do not agree on the sign changes. Large decreases are projected for the driest chains (up to -75 % in the South) while large to very large increases are projected for the most humid ones (up to 75 % in the North). We refer to Tramblay et al. (2025) for further analyses of 285 projected high flows from the Explore2 dataset.







**Figure 4.** 5 %, 50 %, and 95 % quantiles of the climate change responses for low flow (QMNA), mean annual flow (QA), and high flow (QJXA) changes (2071-2099 relative to 1976-2005) for the RCP8.5.





### 3.3 Climate change response uncertainty and internal variability

Figures 5 and 6 compare the CCR uncertainty (CCRU) and the internal variability IV for the climate and hydrological indicators, respectively. Except for summer temperatures, IV is large to very large and often of the same order or even greater than CCR uncertainty.

For seasonal temperature, year-to-year fluctuations around the CCR are of the order of 1 °C (Fig. 5). IV is much lower than CCR uncertainty in summer but similar in winter. Results for ET0 are roughly similar (Fig. S5 in the Supplement). IV is rather large, slightly smaller than CCR uncertainty in summer, slightly larger in winter.

    For precipitation, IV is very large (> 30 %), much larger than CCR uncertainty (between 10 % and 30 %), especially in winter and along the Mediterranean sea (>50 %, Fig. 5). For maximum annual precipitation, IV is large (often greater than

40 %, see Fig. S5 in the Supplement). It is also much larger than CCR uncertainty (10 %-20 %).

    For the three hydrological indicators, IV is logically also very large (Fig. 6). For annual flows, IV is larger than 30-40 % in many regions. For low and high flows, it is higher than 40 % almost everywhere (higher than 100 % for low flows locally). It is often more than two times larger than CCR uncertainty.

### 3.4 Projections for lower emissions

Figures S1 and S2 in the Supplement show the 5 %, 50 %, and 95 % quantiles of the CCR projected for the climate indicators, for the lower emission scenarios RCP2.6 and RCP4.5 respectively. For temperature and related indicators, changes projected with lower emission scenarios are generally smaller than those under RCP8.5, while exhibiting similar spatial patterns. In contrast, projections for precipitation and hydrological indicators do not necessarily lead to the same conclusions. For these indicators, inter-model spread remains substantial, with frequent disagreement among models regarding the sign of change.

Moreover, the direction of projected changes does not always align with those under RCP8.5. For instance, under RCP2.6, winter precipitation is projected to increase in the South-East, and some model chains also indicate increases in summer.

    This pattern is also observed in hydrological projections. Changes projected for lower emission scenarios are not necessarily smaller than changes projected for RCP8.5 (Fig. S3 and S4 in the Supplement). The mean annual flows are projected to increase everywhere in RCP2.6 and almost everywhere in RCP4.5. Low flows are still projected to decrease (much less) for RCP4.5 but

they are projected to increase for RCP2.6, whatever the region. Projected changes for high flows are roughly the same for all scenarios.

### 4   Main uncertainty sources

This section provides key messages about the main sources of uncertainty. They are not the same for the different climate and hydrological indicators. Furthermore, they vary regionally, and uncertainties in projected hydrological changes can be

interpreted according to the corresponding hydroclimatic regimes.



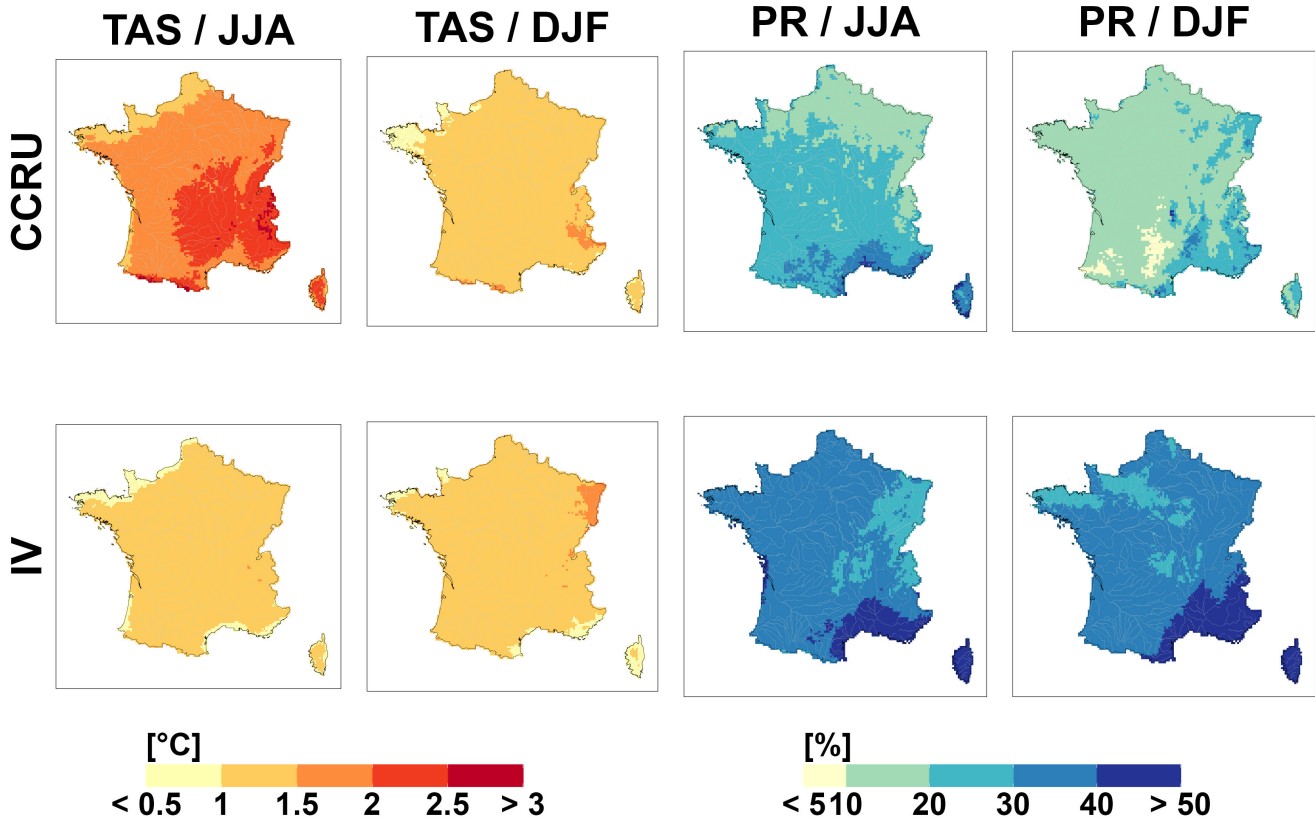

**Figure 5.** Uncertainty components for summer and winter temperature and precipitation changes (2071-2099 relative to 1976-2005) for all emission scenarios. Climate change response uncertainty (CCRU) and internal variability IV (standard deviations expressed in °C for temperature changes and % for precipitation changes).

## 4.1 Main uncertainty sources depend on the climate indicator

Figure 7 shows the main uncertainty components for seasonal temperature and precipitation changes and confirms the results obtained for Europe (e.g. Christensen and Kjellström, 2020; Evin et al., 2021). The relative contributions of the different uncertainty sources to CCR uncertainty depend on the indicator and region.

For temperature projections, scenario uncertainty is by far the main contribution for all seasons. A non-negligible contribution of climate model uncertainty is found in summer. It comes from GCMs for coastal areas (Atlantic and Channel) and from RCMs for mountainous areas and in the East.

For summer precipitation, the main contributions come from RCMs and then from scenarios (between 20 % and 40 %). In winter, they mostly come from GCMs or from RCMs in mountainous areas and in the Mediterranean region. Scenario
uncertainty has a moderate contribution in the North-East. For maximum precipitation, the main contributions come from GCMs and RCMs (Fig. S5 in the Supplement).




For summer ET0, results closely follow those for summer temperature (Fig. S5). However, the contribution of scenario uncertainty is smaller, whereas the contribution of RCMs becomes much larger ($> 20 - 40\%$). For winter ET0, the contribution of scenario uncertainty is smaller to the benefit of RCM uncertainty, which reaches up to $40 - 60\%$ in the West.

With the exception of temperature, the contribution of residual variability to CCR uncertainty is moderate to important, which means that a moderate to large part of the CCR uncertainty cannot be explained by the additive effects of scenario, GCM or RCM uncertainty. This is the case for precipitation in both seasons, for instance. In summer, the unexplained fraction of CCR uncertainty varies from 10-20 % in the West to 20-40 % in the East. In winter, it varies from 20-40 % in the East to 40-60 % in the South-East. Evin et al. (2021) show that interactions between scenarios and climate models can be important

in France for winter precipitation (see their Figure S7). In particular, scenario/GCM and GCM/RCM interactions can partly explain this important residual variability in the current study.

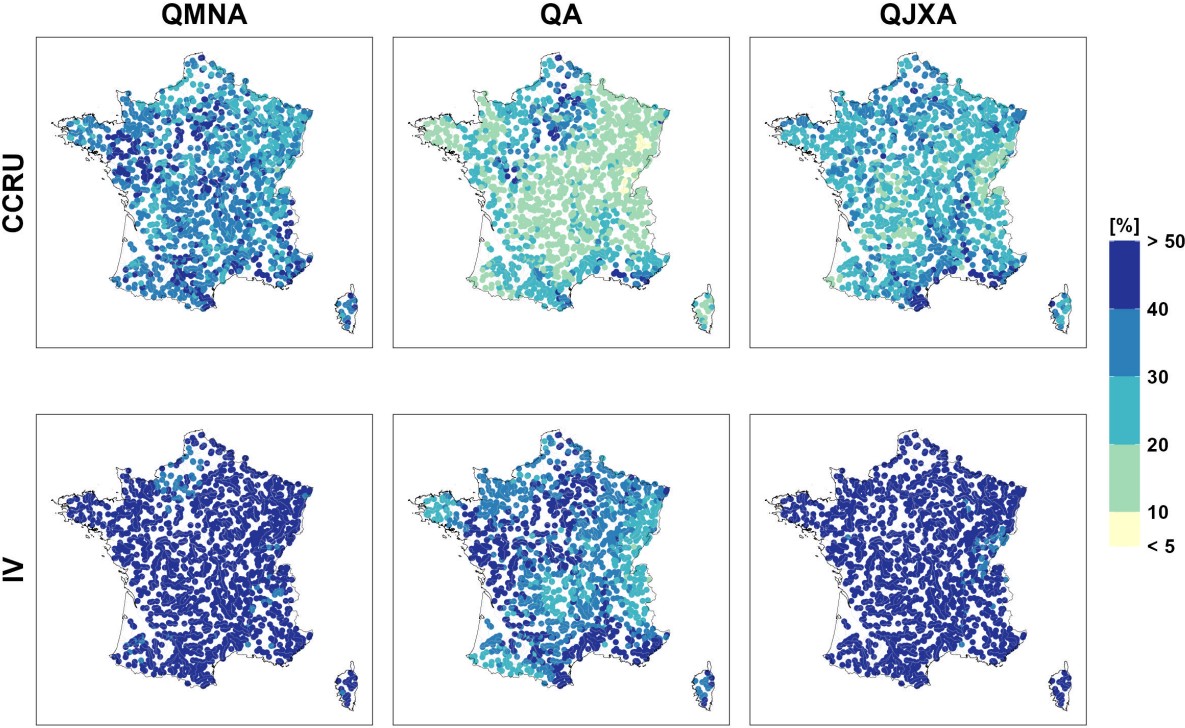

**Figure 6.** Uncertainty components for low flow (QMNA), mean annual flow (QA), and high flow (QJXA) changes (2071-2099 relative to 1976-2005) for all emission scenarios. Climate change response uncertainty (CCRU) and internal variability IV (standard deviations expressed in % for hydrological changes).





**Figure 7.** Uncertainty components for summer and winter temperature and precipitation changes (2071-2099 relative to 1976-2005). Percentage contribution of uncertainty sources (scenario, GCM, RCM, residual variability RV) to the CCR variance (CCRV).




## 4.2 Main uncertainty sources depend on the hydrological indicator

Figure 8 presents the main uncertainty components for hydrological changes. These results are generally consistent with previous studies (e.g. Lafaysse et al., 2014; Chegwidden et al., 2019; Lemaitre-Basset et al., 2021; Aitken et al., 2023). The extent to which each uncertainty source contributes to the spread in projected changes also depends on the region and indicator.

For low flows, CCR uncertainty comes from emission scenarios, especially in southern France (>40-60 %), from RCMs, especially in the East (>20-40 %), and to a lesser extent from HMs in the North-West and in the Alps (>20-40 %). The uncertainty from GCMs is significantly lower than that from RCMs. However, it remains notable in the northern regions, at the expense of RCP uncertainty, especially in the energy-limited areas. In France, except in mountainous areas, low flows occur in summer when snow and soil moisture are depleted. They are mainly driven by soil processes, evapotranspiration, and sometimes groundwater support. Low flows are thus mainly controlled by the climate conditions that determine the summer water budget and by the HM choice.

For mean annual flows, CCR uncertainty mostly comes from RCMs, especially in the East (>40-60 %), from GCMs (everywhere except in the South-East), and from emission scenarios in the southern half of France (>20-40 %). A non-negligible contribution of HMs is obtained in the Paris Basin region (>10-20 %).

For high flows, CCR uncertainty is predominantly attributable to climate models, with GCMs and RCMs contributing to a comparable extent. This substantial contribution can partly be explained by the considerable uncertainty in projections of extreme precipitation (Fig. 3), but may also reflect uncertainties in other climatic variables. High flows are driven by a combination of factors, including the magnitude and spatiotemporal distribution of precipitation, as well as antecedent hydrological conditions resulting from preceding precipitation events (Tarasova et al., 2023). Using the GRSD hydrological model, Tramblay et al. (2025) demonstrate that projected changes in soil saturation exert an influence on high flows comparable to that of projected changes in intense rainfall. In the current study, the pronounced sensitivity to hydrological model selection in eastern France further supports the role of antecedent conditions in shaping high flow responses.

The unexplained fraction of CCR uncertainty can also be large for hydrological indicators. It varies from 10-20 % in the South to 20-40 % in the North for low flows, mostly between 20-40 % for mean annual flows, from 20-40 % to more than 60 % in a number of sites for high flows (Fig. 8).





**Figure 8.** Uncertainty components for low flow (QMNA), mean annual flow (QA), and high flow (QJXA) changes (2071-2099 relative to 1976-2005). Percentage contribution of uncertainty sources (scenario, GCM, RCM, HM and residual variability RV) to the CCR variance (CCRV).



## 4.3 Main uncertainty sources by hydroclimatic regimes

The hydrology of French rivers is expected to undergo marked changes in the coming decades, but the high uncertainty surrounding future precipitation makes the nature of these changes mostly unpredictable. However, the projected increase in temperature is robust and, as a consequence, some changes are very likely, especially for some specific hydroclimatic regimes. As already shown by previous studies in the area (e.g. Lafaysse et al., 2014; Vidal et al., 2016; Dayon et al., 2018; Huang et al., 2022) and by additional Explore2 analyses (Sauquet et al., 2024; Strohmenger et al., 2025), warming will deeply modify the hydrological regimes where discharges are significantly influenced by snow accumulation and melt. Snow-dominated regimes will shift to mixed regimes, while mixed regimes will become predominantly rainfall-driven. As a result, snowmelt floods will occur earlier in spring and be less intense.

Figure 9 aggregates the different results shown in this study by hydrological regime (obtained for the reference period 1976-2005). As indicated above, median CCRs (Fig. 9A) highlight more pronounced changes for low flows than for mean and high flows. For low flows, strong decreases are obtained across most hydrological regimes (around -40%) and scenario uncertainty emerges as the dominant source of variability (Fig. 9B, first row), primarily due to the strong dependence of evapotranspiration losses on temperature, as previously discussed. There are two exceptions. The first concerns catchments with a non-contrasted pluvial (NCP) regime, where low flows are likely to remain supported by deep aquifer contributions regardless of the emission scenario. In this case, the main source of uncertainty arises from climate models, but hydrological models (HMs) also contribute due to their varying representations of deep groundwater storage. The second exception involves catchments with a nival (N) regime, where low flows typically occur in winter due to a high proportion of solid precipitation during this season (Fig. 1C). Under warmer conditions, this proportion is expected to decrease, leading to increased winter discharges. As HMs vary in their representation of snow processes, HM uncertainty plays a particularly significant role in this case.

For mean annual and high flows, the combined uncertainty from GCMs and RCMs is relatively consistent across all regimes. In contrast, the contributions of scenario and HM uncertainties vary by regime. For mean annual discharges, scenario uncertainty is minor in rainfall-dominated regimes but more significant in snow-dominated ones. HM uncertainty is generally low, except in catchments where low flows make up a substantial portion of the mean annual flow (i.e., P and PNC regimes). For high flows, scenario uncertainty is negligible across all regimes. HM uncertainty, while also similar across regimes, is more pronounced due to differing representations of runoff generation in the models—particularly in snow-dominated regimes, where high flows are primarily driven by snowmelt.

A qualitative comparison between Figures 1B, 4 and 8 shows that projected changes are only partially influenced by whether catchments are water- or energy-limited. In the South, the decrease of mean annual flows, which highly depends on the interplay between precipitation and evapotranspiration, tends to be larger in water-limited areas. In the North, mean annual flows tend to decline in water-limited catchments, while they increase in energy-limited ones. However, projected changes in high and low flows appear largely independent of the hydroclimatic regime. For example, the significant reduction in low flows projected for the South is widespread across different regimes. A notable exception is the case of high flows in the South, which tend to increase more in water-limited areas. Across most locations and hydrological variables, the sources of uncertainty contributing



to CCR uncertainty are quite similar between water- and energy-limited regimes, especially regarding HM uncertainty. An exception occurs in the energy-limited catchments of the North, where HM uncertainty is greater for projections of both low and mean annual flows compared to other regions.

**Figure 9.** Results for the different hydrological regimes. A) Median CCR (2071-2099 relative to 1976-2005) for low flow (QMNA), mean annual flow (QA), and high flow (QJXA) changes for the RCP8.5 and B) Percentage contributions of uncertainty sources to CCR variance (CCRV). Rainfall-dominated regimes: highly contrasted pluvial (HCP), contrasted pluvial (CP), pluvial (P) and non-contrasted pluvial (NCP); snow-dominated regimes: Nival (N), Nivo-Pluvial (NP), Pluvial-Nival (PN).





## 5 Discussion

### 5.1 IV is substantial and should not be overlooked

Our results show that IV can be a substantial source of uncertainty in hydrological projections and are in agreement with most other analyses where IV has been considered (e.g. Lafaysse et al., 2014; Vidal et al., 2016; Chegwidden et al., 2019; Alder and Hostetler, 2019; Ye et al., 2024). The influence of IV is strong when precipitation is a key driver and is particularly pronounced for low and high flow projections (Fig. 6). For the three variables considered here, IV is larger than CCR uncertainty at the end of the century, much larger for less far projection periods (not shown).

Our results show that IV should not be overlooked or disregarded in hydrological impact studies. IV expresses the range of possible future realizations around the climate response. IV is likely to cause substantial deviations from the climate response, potentially leading to unusual or critical years or sequences of years. Disregarding IV could be detrimental for interpretations and use of projections. It could prevent a fair evaluation of the resilience and robustness of natural and anthropogenic systems to climate change, variability, and extremes (Doss-Gollin et al., 2019; Bonnet et al., 2020).

### 5.2 A single or a few individual models can have a large contribution to model uncertainty

As shown in Evin et al. (2021) for Europe, a large contribution of GCM (resp. RCM) uncertainty to CCR uncertainty is often due to a single or a small number of individual GCM models (resp. RCM models). This can be easily identified from the maps of the main effects obtained for each individual model. For summer temperature, for instance (Fig. 10), the large RCM contribution comes mainly from the higher warming signature of HadREM3-GA7-05 and the lower warming signature of WRF381P. For winter precipitation (Fig. 11), the important GCM uncertainty comes mainly from the 'wet signature' of IPSL-CM5A-MR; the large RCM contribution in mountainous areas comes mainly from the large differences between four RCMs (RACMO22E and RCA4 are much drier than HIRHAM5 and WRF381P).

A large contribution of a specific climate model to the CCR uncertainty in climate projections frequently leads to similar contributions for other climate or hydrological indicators. For example, the deviating signals produced by HadREM3-GA7-05 and WRF381P for projected changes of summer temperature are reflected in similar divergent patterns for changes in summer ET0 (Fig. S8 in the Supplement) and low flows (Fig. S12 in the Supplement).

A single or a small number of HMs can also lead to a large contribution of HM uncertainty to CCR uncertainty. As shown by the maps of the main effects obtained for each individual HM (Fig. 12), the important HM uncertainty obtained for mean annual flows in the Paris Basin region (lowland area around Paris) is mainly due to the 'more humid' (or 'less dry') signature of ORCHIDEE. For high flows, the important HM uncertainty in the North-East is mainly due to CTRIP. HM main effects vary more regionally (or by regime) for low flows than for mean and high flows. Higher uncertainty for low flows might be expected where more complex processes are involved (e.g. snowmelt for the nival regime, groundwater-river exchange for the non-contrasted pluvial regime) and are represented differently by the different HMs. High flows are mostly driven by precipitation and are more easily represented by HMs when similar antecedent conditions are simulated.




Note that the main effects of the different model categories often have different spatial scales. Spatial patterns of the main effects of GCMs are mostly large scale, presenting smooth variations whereas those of RCMs and those of HMs, often conditioned by local topography for RCMs or by some specific physiographic features for HMs, are more local, often depicting rough variations. The spatial variation of these main effects can be related to the effective resolution of the corresponding

models, which is often larger than the resolution provided by the simulations (Klaver et al., 2020).

Models with very large main effects compared to the other ones can thus be identified using ANOVA approaches. These deviating models inflate the corresponding uncertainty component. As large uncertainties may dampen important information about future changes, it is important to look carefully at these deviating models and understand the reasons behind the atypical behavior. If the models are deviating for wrong reasons (numerical artifacts, bugs, model inadequacy), a reasonable choice can

be to discard them from the MME. Large deviations by a single model can also indicate that other models are overlooking or oversimplifying a key process that could become critical under future climate conditions. This raises concerns about model transferability, particularly for models based on empirical or highly parameterized approaches.




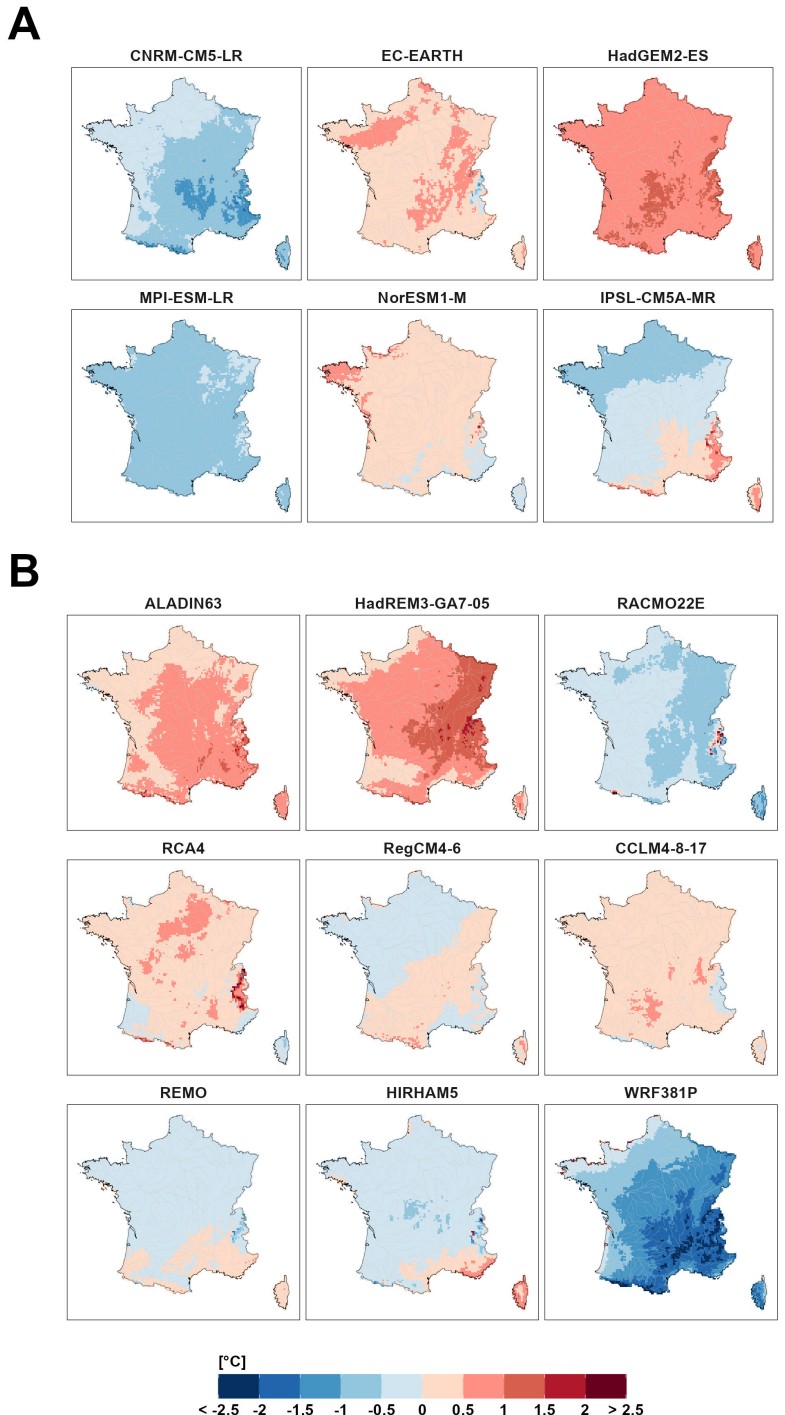

**Figure 10.** Main effects (i.e. deviations from the ensemble mean) of individual climate models for summer temperature changes (2071-2099 relative to 1976-2005). (A) GCMS. (B) RCMs.



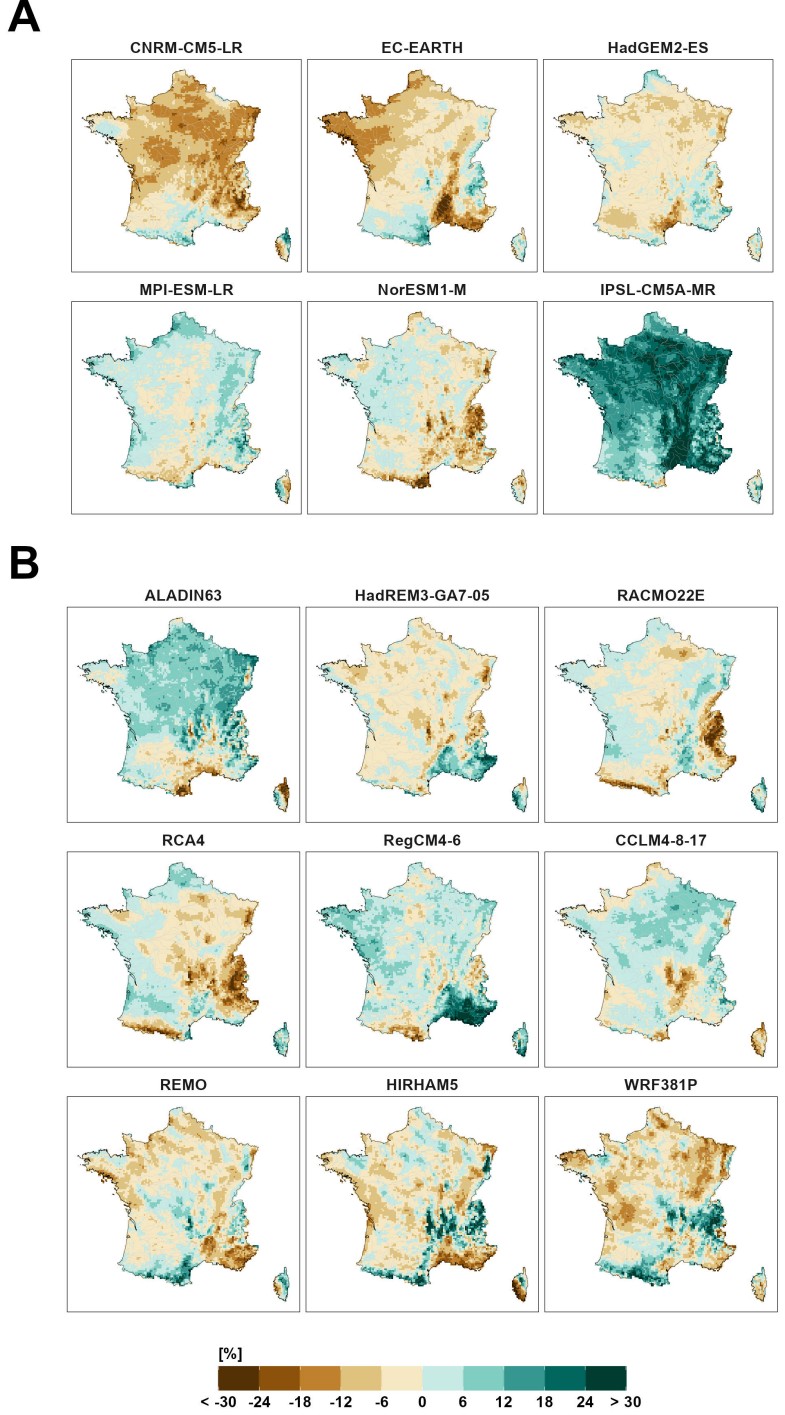

**Figure 11.** Main effects of individual climate models for winter precipitation changes (2071-2099 relative to 1976-2005). (A) GCMS. (B) RCMs.





**Figure 12.** Main effects of individual HMs (CTRIP, GRSD, ORCHIDEE and SMASH) for low flow (QMNA), mean annual flow (QA), and high flow (QJXA) changes (2071-2099 relative to 1976-2005).



## 5.3 Uncertainty contributions depend on projection lead time

CCRs and related uncertainties depend on the future period. Results obtained for other future periods, specifically 2020-
2050 (near future) and 2040-2070 (mid-century), are presented in Evin et al. (2024). As the analysis focuses on the CCR of projections, the dispersion between the main effects of the different models of each category is almost zero for the near future and increases with projection lead time, for all emission scenarios. The uncertainty of each model category and in turn total CCR uncertainty thus also increases with projection lead time. As a consequence, CCR uncertainty tends to be much lower than IV in the near future.

To highlight how results vary over time throughout the 21$^{st}$ century, a summary sheet was created for all stations included in the project, detailing projected changes and associated uncertainties. This summary sheet facilitates the identification of key sources of uncertainty, notably including substantial contributions from individual models to the overall model-related uncertainty. Furthermore, it provides a comprehensive depiction of the range of potential future projections, capturing both systematic changes in climate response and stochastic variations arising from internal climate variability. For illustrative pur-
poses, the summary sheet developed for the Seine catchment at Bazoches-Lès-Bray is presented in Figure 13. It summarizes different results shown above, notably:

- **Panels A and B:** A decrease of lows flows and an increase of high flows with the scenario RCP8.5 (panel A) with an overall agreement of the MME (80% of the CCR with the same sign for the projected changes).

- **Panel C:** A dominant contribution of IV for the total uncertainty (defined a the sum of CCR uncertainty and IV).

- **Panels D-F:** Individual models can have a large contribution to the corresponding component of the CCR uncertainty. For example, the GCM IPSL-CM5A-MR leads to higher projected changes for high flows (panel D, last column), the HM ORCHIDEE has a 'more humid' (or 'less dry') signature for mean annual flows and the HM CTRIP has a 'more dry' (or 'less humid') signature for high flows (panel F).






**Figure 13.** Climate Change Responses and Uncertainties in Explore2 projections of low flow (QMNA), mean annual flow (QA), and high flow (QJXA) changes for the Seine catchment at Bazoches-Lès-Bray. (A) Relative changes (%) for each scenario: median CCR (colored line), dispersion (quantiles 5 % and 95 %) of the CCR (central colored band), envelop curves of interannual fluctuations around the CCR due to Internal Variability (light colored band). (B) Agreement on the sign of the projected change. Triangles point upwards (resp. downwards) if more than 80 % of the CCR are positive (resp. negative). (C) Fraction of total uncertainty $CCRV(t) + IV$ (in variance) explained by the different sources of uncertainty. (D-F) Main effects of GCMs, RCMs and HMs.



### 5.4 Some model limitations are to be acknowledged

Explore2 projections are obviously not free of limitations, partly due to model imperfections. Some known limitations are reported here.

#### 5.4.1 Climate models

The compatibility between a RCM and its forcing GCM is not always guaranteed (McSweeney et al., 2015). GCM/RCM discrepancies exist in the large-scale EUROCORDEX projections for summer, mainly due to differences in the representation

of aerosols (Boé et al., 2020) and atmospheric physics (Taranu et al., 2023). The consequential effect on solar radiation may impact the reference evapotranspiration ET0 used in our study, in particular through its effect on surface temperature (Schumacher et al., 2024). In Explore2, this non-consistent radiation effect was clearly identified for some GCM/RCM combinations (Ribes et al., 2022; Marson et al., 2024) that were thus discarded from the Explore2 ensemble.

#### 5.4.2 Bias Adjustment Models

In Explore2, climate projections were adjusted with ADAMONT (Verfaillie et al., 2017) and CDF-t (Michelangeli et al., 2009) methods. For consistency and simplicity, this study focuses exclusively on ADAMONT-adjusted projections. On the one hand, adjusted projections from CDF-t are only available at a daily time scale and cannot be used to force HMs with hourly meteorological inputs (e.g., ORCHIDEE, CTRIP). On the other hand, the contribution of BAM uncertainty to CCR uncertainty for temperature and precipitation was found to be much smaller than that of the other sources of uncertainty (Evin et al., 2024).

Compared to other studies (e.g Alder and Hostetler, 2019; Senatore et al., 2022; Lafferty and Sriver, 2023), uncertainty due to BAMs is likely underestimated in Explore2 as only two similar approaches relying on quantile mapping were considered. Two potentially critical issues associated with BAMs are also to be reported as they potentially influence the realism of some features of projected scenarios. They come from the strong hypotheses respectively required 1) for the adjustment of high and extreme precipitation values and 2) for the application of the methods in modified climates (transferability assumption).

For example, the adjustment applied by ADAMONT is conditional on the weather regimes from the GCMs, which are not necessarily those of the RCMs (Boé et al., 2020).

#### 5.4.3 Hydrological Models

Similarly, HM uncertainty might be underestimated in this study. Including HMs tends to increase the spread of hydroclimate projections, as shown in Explore2 results for French sub-regions using 6–9 HMs (e.g., Evin et al., 2024, for the Loire basin).

HMs provide a simplified representation of catchment characteristics and hydrological processes, which vary widely in space and time. A broad range of models exists globally, offering different perspectives on future hydrology. Our findings highlight the importance of using a diverse set of HMs to better capture this range. However, the optimal type and number of HMs likely depend on the hydroclimatic context and may require as much diversity as climate models.



Model diversity does not guarantee the relevance of hydroclimate projections and must also represent a large variety of hydrological processes that are accounted for. In Explore2 for instance, HMs are not really suitable for catchments with important glacier coverage (found in some high-elevation alpine catchments) or with strong surface-groundwater interactions, notably in the case of karstic basins. The transferability of HMs in a modified climate context is also not ensured. For instance, with the exception of ORCHIDEE, Explore2 HMs do not account for the $CO_2$ rising effect on vegetation physiology, and in turn on evapotranspiration and land hydrology (Vicente-Serrano et al., 2022; Lemaitre-Basset et al., 2022). Accounting for this effect makes ORCHIDEE simulate lower evapotranspiration losses and higher runoff (not shown). Finally, as indicated in Section 2, streamflow projections from the Explore2 dataset are based on 'natural' hydrology and do not account for human influences such as water management infrastructure, water usage, or land use, all of which will clearly affect future hydrological conditions.

## 6 Conclusion

Explore2 projections reflect the current state of scientific knowledge on climate change and natural hydrology for Metropolitan France. They have been produced for a large number of stations along French rivers with a variety of hydrological models from a large ensemble of bias-adjusted regional climate projections.

We use QUALYPSO to assess how a specific climate or hydrological indicator is expected to change. This method, designed to tackle the main challenges of uncertainty partitioning in multi-model multi-scenario ensembles, leverages the very large Explore2 dataset, based on the rich but incomplete and unbalanced EUROCORDEX ensemble. Although the configuration is complex to work with, QUALYPSO enables the characterization of projected climate change responses (CCRs) across various climate and hydrological indicators. Specifically, it provides insights into the mean and the dispersion of the CCRs obtained from different modeling chains. Moreover, it facilitates the estimation of all sources of uncertainty—scenario, GCM, RCM, and HM uncertainties—as well as potential additional dispersion in future changes arising from internal variability. Lastly, it enables the identification of the main effects associated with each model within each model category, providing a clearer understanding of how individual models compare and where the primary sources of uncertainty arise. The key takeaways from our analysis are as follows.

France is projected to experience warming, with greater temperature increases expected under higher future greenhouse gas emissions. It is also projected to warm more in summer, especially in southern France, and in mountainous areas in winter. Warming will lead to an increase in potential evapotranspiration everywhere, particularly in summer. In contrast, the future patterns of seasonal precipitation are more uncertain. For the RCP8.5, models mostly agree on a summer precipitation decrease in the South and a winter precipitation increase in the North. Annual maxima of daily precipitation are projected to increase, particularly in a large northern half of France. For all precipitation indicators, the inter-model dispersion is large and climate models often disagree on the direction of the projected changes.

The future of French river hydrology is mostly shaped by the projected warming and the uncertain future of precipitation. For hydrological regimes sensitive to temperatures, larger greenhouse gas emissions will lead to larger hydrological changes. For



water-limited regimes in southern France, both annual discharges and low flows are projected to decrease. As shown by Sauquet et al. (2025), snow-dominated catchments will evolve into mixed regimes, while mixed regimes will shift toward rainfall-dominated systems. For almost all catchments, the increased potential for evapotranspiration should increase the severity of low flow periods. This increase should be smaller for catchments with snow-dominated or non-contrasted pluvial regimes. Low flows are expected to increase due to altered snowpack dynamics in the first case, while in the latter, they should remain supported by deep aquifer contributions.

Whatever the climate and/or hydrological indicator, the dispersion of the CCR between modeling chains is large. The main uncertainty sources depend, however, on the indicator. Scenario uncertainty has the greatest impact in snow-dominated regimes—where snowpack dynamics are crucial—and in water-limited regions or seasons, such as Mediterranean areas or during summer low flows, where evapotranspiration losses play a key role. As climate projections are very uncertain, the choice of the climate model is very important for mean annual flow projections, especially for the rainfall-dominated regimes. In contrast to the choice of the GCM, the choice of the RCM also matters for low flows. The choice of the HM can be important. It is high for low flows, moderate for annual discharge, and low but not negligible for high flows.

A significant contribution of GCM (or RCM or HM) uncertainty to CCR uncertainty is often attributed to one or a few individual models within each category. Determining the main effects of each model can offer valuable insights that can guide future model evaluations, improvements, or adjustments.

Explore2 highlights slow and long-term changes in different hydroclimatic indicators. Additionally, natural variability is an inherent characteristic of all hydroclimatic indicators. Our results confirm that IV can be substantial compared to CCR uncertainty. As a result, it may lead to significant departures from the climate response, potentially giving rise to anomalous or critical years or multi-year periods. IV is frequently overlooked in hydrological impact studies. Our results confirm that IV can be large and should be carefully considered to ensure the robustness of adaptation studies.

While Explore2 is the largest ever produced MME of hydrological projections from regional climate projections and at the scale of a country, it must not be over-interpreted and needs to be presented as it truly is. The Explore2 MME is conditional on scenarios and models chosen and available at the time the project was carried out. Models are imperfect representations of real systems and necessitate various simplifications. A probabilistic approach to characterize the ensemble would not be relevant. Despite its size, the Explore2 ensemble may not encompass the exact climate evolution that will occur in the coming decades. The chance to encounter unexpected climate events remains, and the possibility for climate surprise must be acknowledged.

**Author contributions**

GE and BH designed the research. GE analyzed the data and produced the figures presented in this work. GE and BH prepared a draft of the manuscript. Adjusted climate simulations were produced by LC and PM for ADAMONT and by MV and YR for the CDF-t. LH processed Explore2 projections and produced the time series of hydroclimate indicators. AR produced preliminary analyses. SM, LS, GT, PH, AD, FC produced the hydrological simulations with CTRIP, GR4J, ORCHIDEE and SMASH. FHe, CM, ML, JG, JPVe, JB, JMS, FHa produced Explore2 hydrological simulations for subregions not presented



in this work. CM, AR, GE, JPVi and ES interacted with a panel of stakeholders to define the content of summary files. All the authors contributed to interpreting results, discussing findings, and improving the manuscript. ES supervised the project and acquired the funding.

**Code and data availability**

The QUALYPSO approach is implemented in an R package available on CRAN (Evin, 2023). The codes used to produce the
analyses and figures shown in this study are shared on GitHub and are available at https://github.com/guillaumeevin/Explore2/. Summary sheets of the uncertainties are available on the following dataverse repository: https://doi.org/10.57745/3LP5EN (Evin, 2025). The Explore2 dataset is associated with the following digital object identifier https://doi.org/10.57745/YHMBHC. The hydrological data can be downloaded in netCDF file format through the open platform for French public data dedicated to the Explore2 project (https://entrepot.recherche.data.gouv.fr/dataverse/explore2) and the DRIAS-Eau website (https://www.
drias-eau.fr/). SAFRAN is available at https://meteo.data.gouv.fr/datasets/6569b27598256cc583c917a7.

**Authorship**

We confirm that the manuscript has been read and approved by all named authors, including the order of authors listed.

**Declaration of Competing Interest**

The authors declare no conflict of interest related to the results presented in this paper.

**Acknowledgments**

This research was financed by the Explore2 project with support from the French Biodiversity Agency (OFB) and the French Ministry of Ecological Transition (MTECT). The authors wish to thank Patrick Arnaud, Flora Branger, Yvan Caballero, Sandra Lanini and Charles Perrin for the rich interactions about this topic during the project.





## Appendix A: QUALYPSO

This appendix provides additional details about the key steps of the approach, for an application on an ensemble of hydrological projections or climate projections (in that case, the effect $EH_h(t)$ of the HMs is irrelevant hereafter):

- **Estimation of the climate response $CR_i(t)$ of each projection:** As discussed in Section 2, a key stage in the analysis is the estimation of the climate response for each projection $i$. We used a trend model based on cubic splines applied to the transient projections available over the entire simulation period. For each projection, the smooth trend predicted by 585 the cubic spline corresponds to the climate response $CR_i(t)$.

- **Estimation of the climate change response $CCR_i(t)$ of each projection:** In line with current practice, absolute changes are considered for temperature (Eq. A1) and relative changes for the other indicators (Eq. A2). These changes are relative to the reference period (1976-2005), using the climate response estimated for its central year (i.e. $t_{ref} = 1990$):

$$CCR_i(t) = CR_i(t) - CR_i(t_{ref}), \tag{A1}$$

$$CCR_i(t) = \frac{CR_i(t) - CR_i(t_{ref})}{CR_i(t_{ref})}, \tag{A2}$$

with $t$ the central year of the future period considered, $t_{ref}$ the central year of the reference period, $CR_i$ the climate response of the projection $i$ and $CCR_i$ the corresponding climate change response.

- **Estimation of the internal variability:** We assumed that the internal variability of a projection is constant over time (in variance). For each projection $i$ and each time horizon $t$, we first consider the differences between the raw projection 595 $Y_i(t)$ and the climate response $CR_i(t)$ as follows:

$$D_i^*(t) = Y_i(t) - CR_i(t), \tag{A3}$$

$$D_i^*(t) = \frac{Y_i(t) - CR_i(t)}{CR_i(t_{ref})}. \tag{A4}$$

depending on whether we are considering absolute or relative changes, respectively. Internal variability is then estimated by the time variance of these differences $D_i^*(t)$, throughout the period under consideration:

$$IV_i = \mathbb{Var}\{D_i^*(t)\}. \tag{A5}$$

- **Estimation of the ensemble mean and the main effect of each scenario/model:** For a given emission scenario $s$, the climate change response $CCR_i(t)$ of a projection $i$ obtained with a simulation chain composed of a GCM $g$, an RCM $r$, and an HM $h$ is assumed to be the sum of the individual main effects, as follows:

$$CCR_i(t) = M(t) + ES_s(t) + EG_g(t) + ER_r(t) + EH_h(t) + \epsilon_{s,g,r,h}(t), \tag{A6}$$

where $M(t)$ is the ensemble mean for a future period $t$, and $ES_s(t), EG_g(t), ER_r(t), EH_h(t)$ are respectively the main effect of the RCP scenario $s$, the GCM $g$, the RCM $r$, and the HM $h$ and where $\epsilon_{s,g,r,h}(t)$ is the residual term, i.e. the




part of the CCR that is not explained by the sum of the main individual effects. This estimation is carried out for each future time horizon $t$ using a linear regression model, where the least squares of the residuals $\epsilon_{s,g,r,h}(t)$ are minimized.

In a complete ensemble of projections, the main effect of a model is easily interpreted as the mean difference between 1) the CCR of all the projections using this model and 2) the mean CCR of the ensemble. By construction, the sum of the main effects of the different models belonging to a given category of models is zero (Eq. A7):

$$\sum_s ES_s(t) = \sum_g EG_g(t) = \sum_r ER_r(t) = \sum_h EH_h(t) = 0. \tag{A7}$$

– **Estimation of the different sources of uncertainty:** The internal variability of the ensemble is estimated by the average standard deviation of the fluctuations around the CCR, i.e. the square root of the ensemble mean of the internal variability variances estimated for the $N$ projections of the ensemble:

$$IV = \sqrt{\frac{1}{N} * \sum_i^N IV_i}. \tag{A8}$$

For a future time horizon $t$, the uncertainty associated with the emission scenarios and the uncertainties associated with each model category (GCM, RCM, and HM) corresponds to the variance of the corresponding main effects. For example, GCM uncertainty is estimated by the variance of the main effects $EG_g(t)$, $g = 1..N_g$, estimated for the $N_g$ GCMs considered. Finally, the residual variability $RV(t)$ is the variance of the residuals $\epsilon_{s,g,r,h}(t)$.

The total uncertainty variance CCRV(t) of the CCR is obtained as the sum of the residual variability and the variance of each model category:

$$CCRV(t) = \mathbb{V}\mathrm{ar}(ES_s(t)) + \mathbb{V}\mathrm{ar}(EG_g(t)) + \mathbb{V}\mathrm{ar}(ER_r(t)) + \mathbb{V}\mathrm{ar}(EH_h(t)) + RV(t), \tag{A9}$$

and $CCRU(t) = \sqrt{CCRV(t)}$ is the corresponding standard deviation. These different estimates can be used to estimate the fraction of the total uncertainty resulting from each source of uncertainty:

$$FV_x(t) = \mathbb{V}\mathrm{ar}_x(t)/CCRV(t), \tag{A10}$$

where $FV_x(t)$ is the fraction of the total uncertainty explained by the source of uncertainty associated with $x$ (here $x$ = scenario, GCM, RCM, HM, or RV).



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
