# Peer review of "Uncertainty sources in a large ensemble of hydrological projections: Regional Climate Models and Internal Variability matter"

_EGUsphere, 2025_

## Author Comment (AC1)

**Community comment 1 Rasmus Benestadt**

**CC1.1. I think this paper is very interesting and a welcome contribution. I also appreciate the opportunity to discuss some of the points made herein.**

Thanks for this positive feedback as well as the open discussion on different points mentioned below. They have been taken into account in the revised version of the manuscript.

**CC1.2. One point raised is "Model uncertainty arises from model imperfections" which is important, but this paper neglects uncertainties connected with the downscaling approach because it fails to mention others then dynamical downscaling (aka regional climate models, abbreviated as 'RCMs'). There are also other ways of downscaling global climate models (GCMs) which are based on entirely different assumptions and come with different strengths and weaknesses. We expect them to produce similar results if they all are skillful, independently of each other. Hence, if dynamical and empirical-statistical downscaling give similar outlooks, then the results can be considered as being more robust. Therefore, I recommend that the paper includes some discussion on empirical-statistical downscaling in order to get a more complete picture on uncertainties associated with modelling.**

We agree that it would be worth mentioning empirical-statistical downscaling approaches in the discussion, in particular in the subsection 5.4.1. which mention the limitations of the climate models. A paragraph will be added to discuss alternative downscaling approaches, including empirical-statistical methods and recent artificial intelligence approaches.

**CC1.3. The need of bias-adjustment also introduces uncertainties. It's in a fashion similar to 'sweeping the problem under the carpet', but also it assumes that the present biases are similar to those in a changed climate.**

There is at least one example of downscaling precipitation statistics large multi-model CMIP ensembles that may be of relevance: https://doi.org/10.5194/hess-29-45-2025. However, this example focuses on downscaling daily precipitation statistics and may require an additional step using weather generators to produce time series needed as input for hydrological models. On the other hand, the downscaled precipitation statistics provides a rule-of-thumb estimate for number of days per year with heavy rainfall.  The method described in this paper will provide a basis for studying the connection between climate change and water-born diarrhoea outbreak in the EU-SPRING project (https://www.springsproject.eu/).

One motivation for downscaling statistical properties (e.g. parameters of statistical distributions) is that statistical properties often are easier to predict/quantify than individual outcomes.

In some cases, climate internal variability (IV) actually provides some useful information about inter-annual variability and the range of plausible outcomes. For example, downscaled results of large ensembles provide a band of plausible temperatures in https://doi.org/10.1073/pnas.2503806122 that can be compared with historical temperatures, and such an evaluation reveals whether the downscaled results match the observed inter-annual variability. The mean of the model spread can be interpreted as the climate normal, whereas upper and lower limits represent hot and cold years. It is also interesting to note that the ensemble spread in some cases is close to being normally distributed.

We agree that statistical downscaling methods can be a valuable approach for producing statistical properties of meteorological variables under climate change. Applying weather generators to produce continuous time series based on these indices could be an option. Such approaches are actually under consideration in the context of the project Explore2 and are under development.

Large multi-model ensembles (with multiple runs) are obviously of interest to better estimate the expected mean and variability of present (and future) climates. Often, these large MMEs are not considered for hydrological impact studies because they are much too computational demanding. Note that as soon as transient climate projections are available, climate responses can be estimated from a single run, and the uncertainty assessment can be carried out even when multiple runs MMEs are not available, as is the case for most impact studies (Hingray et al. 2019). This is also illustrated in our work with QUALYPSO. Note that this "one member limitation issue", typically found in impact studies, is mentioned in the introduction.

**CC1.4. The statement "To our knowledge, the Explore2 MME is the largest ensemble of hydrological projections ever produced from regional climate experiments at the scale of a country" is probably true - see https://doi.org/10.5194/hess-29-45-2025 where MMEs were downscaled for SSP370, SSP126, SSP245, and SSP585, each with ~30 ensemble members (there are also unpublished results (work in progress) with downscaling total annual precipitation of 200-300 ensembles for each SSP).**

When it comes to evaluation, it is not clear if the results are evaluated involving the complete chain of models. I.e. is the downscaling combined with hydrological modelling of GCM historical runs able to reproduce observed trends and inter-annual variability? Also,  are the RCMs able to reproduce past variability and trends?

The evaluation issue of climate models is indeed a very critical issue. It is also a difficult one and requires it to be carefully addressed. This issue is out of scope of the manuscript and we could not give him the full place it should deserve. As mentioned in the manuscript, we refer to the studies led by the EUROCORDEX community for the evaluation of the climate simulations (e.g. Coppola et al., 2020). For the evaluation of the hydrological MME itself, we refer to https://doi.org/10.5194/egusphere-2025-1788.

---

## Author Response (AR1)

**Editor decision**

**EC1.1. The manuscript has now been evaluated in light of the referees' reports and your replies. All three referees agree that the study is scientifically sound, well written, and makes a valuable contribution to the understanding of uncertainty propagation in climate–hydrology modeling chains. The motivation, structure, and scope of the work are particularly appreciated, as is the careful interpretation of results across contrasting hydroclimatic regimes. The reviewers' main concerns focused on (i) clarification of several methodological aspects related to trend smoothing, internal variability estimation, and variance decomposition, and (ii) the interpretation and practical implications of cases where individual models or model chains contribute disproportionately to overall uncertainty. Based on your responses, these points have been addressed satisfactorily, notably through additional explanations and clarifications provided in the author replies and supplementary material.**

We thank the editor for this positive feedback.

**EC1.2. 1. Methodological clarity: Please ensure that the explanations regarding the use of cubic spline smoothing, the definition and role of $CR_i(t)$, the estimation of internal variability, and the construction of $ES_i(t)$ are clearly and concisely integrated into the main text and/or appendices, so that readers can fully understand the rationale and implications without relying on the response documents alone. Where relevant, please briefly clarify the suitability of the smoothing approach for precipitation and extreme hydrological indicators.**

Some clarifications have been added to section 2.3 and in the Appendix. We believe that the new illustration provided in the Appendix of the revised version will also help to understand the differences between the different quantities. Concerning the suitability of the smoothing approach, we have added a comment in the new paragraph 5.4.4.

2. Notation and definitions: Please double-check that the definitions and notation for IV, RV, and FV are fully consistent and unambiguous throughout the manuscript, and clarify their relationships where needed.

Definition and notations have been checked and clarified if needed. Again, the illustration will probably help to understand the difference between IV, RV, and FV.

3. Interpretation of discordant models
While I acknowledge that a full diagnostic analysis is beyond the scope of the present paper, please slightly strengthen the Discussion to more explicitly guide readers on how the identification of strongly contributing or discordant models could be used in practice.

A paragraph has been added at the end of Section 5.2. to answer this point.

**Reviewer #1**

**RC1.1. This study explored the sources of uncertainty in different components of the model chain and investigated their contributions to two climate indicators and three hydrological indicators. The variation in performance was evaluated based on different regional characteristics. I consider the motivation of this paper very good, especially in the context of using ensembles for climate projection. The structure of the paper is well organized, and the presentation is good as well. However, I have a few concerns about the calculation methods that need to be resolved before the paper can be accepted.**

We thank the reviewer for the overall positive comment. We understand these concerns and we have proposed some modifications in the revised version. Note that most of these points have been addressed in previous articles, in particular the methodological paper Evin et al., 2019, https://doi.org/10.1175/JCLI-D-18-0606.1 and the application of the same methodology to a EUROCORDEX MME (Evin et al., 2021, https://doi.org/10.5194/esd-12-1543-2021). For this manuscript, the aim was 1/ to present the results of the uncertainty analysis for a very large ensemble of hydrological projections where uncertainty come from GCM, RCM, BAM and HM and 2/ to show how such a method allows to better understand where uncertainty models come from (from which model category first, but also from which individual models). We choose to explain the main assumptions of the method in Section 2 and provide the technical details in the Appendix. To answer the reviewer's concerns, we have added a few additional details in Section 2 to clarify some technical points (e.g. estimation) and a new figure has been provided in the Appendix to illustrate the different steps of QUALYPSO. We believe that this new figure is very helpful to understand the differences between the different quantities.

Evin, Guillaume, Benoit Hingray, Juliette Blanchet, Nicolas Eckert, Samuel Morin, and Deborah Verfaillie. « Partitioning Uncertainty Components of an Incomplete Ensemble of Climate Projections Using Data Augmentation ». *Journal of Climate* 32, n° 8 (2019): 2423-40. https://doi.org/10.1175/JCLI-D-18-0606.1.

Evin, Guillaume, Samuel Somot, and Benoit Hingray. « Balanced Estimate and Uncertainty Assessment of European Climate Change Using the Large EURO-CORDEX Regional Climate Model Ensemble ». *Earth System Dynamics* 12, n° 4 (2021): 1543-69. https://doi.org/10.5194/esd-12-1543-2021.

**RC1.2. What is the purpose of applying cubic splines to the projection and what are the effects on the trend analysis (Line 583)? What is the meaning of the smooth trend denoted as CRi(t)? Please elaborate on the calculation method. Additionally, is the smoothing suitable for precipitation and hydrological indicators (especially max1D)?**

The climate response of a simulation chain, denoted as CRi(t), corresponds to the long-term trend of the simulated projection (section 2.3.1). It is assumed to have a temporal variation that is inherently gradual and smooth. In this study, this long-term

trend is estimated using a cubic spline model applied to the corresponding projection available for 1976-2099. As mentioned in the manuscript, other trend functions could be used to extract the climate response of each chain in QUALYPSO (linear trend, polynomial trend, etc.).

The calculation method is described in Section 2.3.2 and will be modified to clarify the reviewer's questions. Cubic smoothing splines are implemented by the function smooth.spline in R. For all indicators except temperature (e.g. seasonal precipitation, annual maxima of daily precipitation, and hydrological indicators), the inter-annual variability is relatively large compared to the long-term trend, so the smoothing parameter spar was set to 1.1 to reduce the model's flexibility. This prevents misattributing the low-frequency fluctuations caused by inter-annual variability to the climate response. For temperature, we apply a lower smoothing parameter value of 1 to provide more flexibility. The choice was defended in previous studies (Evin et al., 2021) and checked by visual inspection of the climate responses for this study. However, we agree that extracting the forced climate response can be difficult for some indicators, for example when they often reach a bound (e.g. zero for positive values) and/or when the interannual variability is large (as is the case for annual precipitation maxima). This point is now discussed in paragraph 5.4.4 in the discussion.

**RC1.3. The authors may need to showcase some results from this step.**

We thank the reviewer for this suggestion. An illustration of the climate responses obtained for one pixel and one catchment has been added to the manuscript in the Appendix (new figure A1).

**RC1.4. In the estimation of internal variability (Lines 593–600), why does the method first estimate Di(t) as the difference between the raw projection (Yi(t)) and CRi(t), rather than directly simulating variability from the raw projection over the target period? Does this step reduce or increase the internal variability? Based on the results, the internal variability is super large—could this be because the smoothing is not applicable?**

As indicated in Section 2.3.1, The high- to mid-frequency fluctuations in the simulated projections result solely from interannual variability. Our approach assumes that it is reasonable to consider a trend model to estimate the climate response of a chain, and in turn fluctuations around, due to interannual variability (deviations from the climate response). Hingray et al. (2019) have shown that this assumption allows providing, for all uncertainty components, more precise estimates than estimates obtained with time-slice approaches. It does not reduce neither it increases the interannual variability but rather separate two components of the total uncertainty: variability of the climate responses and interannual variability. This point is also discussed in the introduction: "*Disentangling the climate response of a given chain from stochastic fluctuations caused by IV is key for a relevant uncertainty analysis. Estimating the climate response can be challenging, particularly for indicators such as precipitation, where IV is significant (Hingray et al., 2019). This difficulty arises because climate outputs blend the climate responses with chaotic fluctuations from IV, which propagate through all the subsequent models in the chain. If for a given GCM multiple members are available and used for subsequent simulations, the climate response of a modeling chain forced by this GCM can be*

*estimated with the multi-member mean of the simulations, and IV can be estimated with the inter-member variability. However, many hydrological studies rely on single-member and time-slice GCM experiments. As a consequence, IV cannot be properly filtered out and, when they are not simply disregarded, stochastic fluctuations from IV are often attributed to GCM uncertainty (see, e.g., Bosshard et al., 2013; Vetter et al., 2017; Gangrade et al., 2020).*" It is true that interannual variability is often large in hydrological impact studies, because precipitation and hydrological indicators are highly variable from one year to the next.

Hingray, Benoit, Juliette Blanchet, Guillaume Evin, and Jean-Philippe Vidal. « Uncertainty Component Estimates in Transient Climate Projections ». *Climate Dynamics* 53, nᵒ 5 (2019): 2501-16. https://doi.org/10.1007/s00382-019-04635-1.

**RC1.5. Please elaborate on the calculation of ESi(t), using one example (e.g., RCP, s). Why is a linear regression model applied, and how is it used (Line 608)?For Equation (A7), does this equation still work if incomplete or unbalanced ensembles are used? How are the effects of incomplete ensembles reflected in the results? Authors failed to explain this in detail since this is the second major question to be solved.**

We thank the reviewer for this comment. The description of the estimation step has been improved in the revised version (paragraph 2.3.4). In short, the individual effects are estimated at once using the linear model A7 which describes a sum of additive terms (Samson et al., 2013). The estimation is implemented by the R function lm using least-squares and standard recipes of numerical linear algebra (QR-decomposition).

Note that in a former application of QUALYPSO, the ANOVA was estimated with a Bayesian approach combined with a data augmentation technique (Evin et al. 2019). Estimates obtained with both approaches are almost identical. Both approaches provide unbiased estimates even when the ensemble is incomplete (see also section 8.1 in Evin et al., 2021). Compared to the regression approach, the Bayesian approach has the advantage of providing the uncertainty of these estimates. However, it is computationally demanding (roughly 100 times more than the regression approach). Estimates using the regression method are also more stable because they do not rely on the sampling of posterior distributions.

Evin, G.; Hingray, B.; Blanchet, J.; Eckert, N.; Morin, S.; Verfaillie, D. Partitioning Uncertainty Components of an Incomplete Ensemble of Climate Projections Using Data Augmentation. *J. Climate* **2019**, *32* (8), 2423–2440.
https://doi.org/10.1175/JCLI-D-18-0606.1.

Evin, G.; Somot, S.; Hingray, B. Balanced Estimate and Uncertainty Assessment of European Climate Change Using the Large EURO-CORDEX Regional Climate Model Ensemble. *Earth System Dynamics* **2021**, *12* (4), 1543–1569.
https://doi.org/10.5194/esd-12-1543-2021.

Sansom, Philip G., David B. Stephenson, Christopher A. T. Ferro, Giuseppe Zappa, et Len Shaffrey. Simple Uncertainty Frameworks for Selecting Weighting Schemes and

Interpreting Multimodel Ensemble Climate Change Experiments. Journal of Climate. 15 juin 2013. https://doi.org/10.1175/JCLI-D-12-00462.1 .

#RC1.8. What is the difference between IV, RV, and FV? Should they use the same definition but with different superscripts/subscripts?

IV is the internal variability, i.e. the standard deviation of the fluctuations around the climate change responses (Eq. A8). RV is the variance of the residuals of the ANOVA model. It corresponds to the unexplained variance of the ANOVA model, i.e the variance of the climate changes responses that can not be explained by the sum of the main effects of the different models (GCM, RCM, HM) considered in the modelling chains. FVis the fraction of total uncertainty variance CCRV(t) resulting from each source of uncertainty (Eq. A10). For a given future time, one FV value is computed for each category of uncertainty source (i.e. for scenario uncertainty, GCM uncertainty, RCM uncertainty, HM uncertainty and RV) and those FV values sum to 1. They are different quantities, with different definitions. As indicated above, we choose to describe these technical details in the Appendix. However, we understand that it might be difficult to understand how all these quantities differ. The illustration of QUALYPSO which has been added in the Appendix should clarify these differences.

#RC1.9. Does the selection of the time span length (i.e., 30 years in this study) affect the results, since a longer time span would likely lead to larger internal variability?

As indicated at l. 591-592, the climate change responses are taken as the absolute or relative differences of the climate responses for the center of the 30-year time period. As the climate response is estimated with a trend model (here a cubic spline), no time span is considered to estimate it. For the sake of simplicity, we refer to 30-year periods but the climate response of a given such period is the value of the trend model for the year at the center of this period. Consequently, the time span does not affect the climate change response. It neither affects the internal variability which is estimated from the annual deviations from the climate response (i.e. the long-term trend).

**Reviewer #2**

**2.1. The manuscript "Uncertainty sources in a large ensemble of hydrological projections: Regional Climate Models and Internal Variability matter" by Evin et al. investigates the outcomes of several climate-to-hydrology modeling chains over the area of France. The perspective is to quantify the uncertainty introduced at any level of the chain by the various combinations of models that can be used at any level of the chain. This study is of great interest because it highlights the value and limitations of future hydrological projections, providing advice on which hydrological indicators can be predicted more robustly and where within the study area.**

We thank the reviewer for this positive feedback.

**2.2. The work is a great effort of synthesis but leaves incomplete a fundamental point that I believe should be extended and integrated in the work. In the Discussion section (and recalled also in the Conclusions) the authors note that, in some cases, some models/combinations of models are discordant, then contributing significantly to the overall uncertainty. They suggest (page 26) to "look carefully at these deviating models and understand the reasons behind the atypical behavior"; then, if the model deviates for "wrong reasons (numerical artifacts, bugs, model inadequacy)" should be discarded, otherwise the analyst should investigate the reason of the dissimilarity further. In this case, the authors acknowledge the possibility that the model is superior to others. This point is extremely important, and the authors also acknowledge it at the end of page 26 and in the Conclusions section.**

However, I would have expected a more thorough investigation of this point, i.e. a deeper analysis of "what to do" in these cases. Clearly, it is not possible to provide general instructions valid for any case, but the study provides a very good base to identify a few specific cases to develop in more detail. For instance, one could look at a specific catchment/climate regime and decide whether one model/model chain should be discarded or, on the contrary, should be preferred because more reliable in a specific context (see e.g. the cases reported in section 5.4.3). I understand that this is not an easy task, but it is necessary to make the paper much more than a descriptive product.

We thank the reviewer for this comment and we agree that this is a crucial point that was a bit eluded in the manuscript. A paragraph has been added in the subsection 5.2 of the revised version. We also agree that a detailed and motivated selection for a specific case would be interesting. However, this is not an easy task and this would probably require an additional section. As the manuscript is already rather long, we leave this detailed illustration for a further analysis.

The recommendation made to the end users of the simulations is to examine summary sheets produced at different levels of aggregation, from local (station level) to regional (hydrographic region level), and make their own choices. In our opinion, it is not possible to recommend a set of models a priori without knowing how the data will be used. Indeed, the choice must be guided by users' needs—for example, whether only

streamflow is required, or streamflow together with other variables, or streamflow at a prescribed set of simulation points available. The choice is constrained by the models' ability to provide all desired variables at the points of interest.

For your information, note that the Explore2 project produced two reports (Sauquet and Héraut, 2023, Sauquet et al., 2025, in French) dedicated to the diagnostic assessments that have been performed for all catchments. They provide an overview of the qualities and performance of surface hydrology and hydrogeological models when driven by the SAFRAN reanalysis. Sauquet and Héraut (2023) also provide recommendations to the end users to help them select or filter some model chains for a specific catchment. Uncertainty assessments shown in the manuscript are thus complemented by a series of thorough diagnostics on the reference period. The purpose of these diagnostics is not to pit the models against one another; rather, it aims to support the selection of a group of models—when several are available—to be prioritized in a prospective exercise. In addition, Héraut et al. (2024) describes some choices which have been to identify atypical simulations: 1. Using the mean annual flow QA on the reference period: a simulation is not plausible if its QA is outside the range [0.5*med(QA), 2*med(QA)], where med(QA) is the multi-model median of all QA values; 2. Using the QA anomaly between the end-of-century QA and the QA for the reference period $\Delta$QA: a simulation is not plausible if $\Delta$QA is outside the range [med($\Delta$QA) +/- 3*$\sigma$($\Delta$QA)], where med($\Delta$QA) and $\sigma$($\Delta$QA) are the multi-model median and standard deviation of all $\Delta$QA values, respectively. If, for one catchment, 50% of the projections corresponding to an hydrological model are not plausible, then, it can be considered that hydrological projections are flawed for this model (imperfect representation of the hydrological processes, wrong drained area) and they are all rejected. This loose set of criteria mainly aims at automatically filtering simulations which exhibit obvious shortcomings.

As an illustration, these recommendations have been followed for a study dedicated to the future water resources of the Isere department, an area located in the French Alps (see report in French: https://www.isere.fr/sites/default/files/2025-05/livrets-methodologiques.pdf). A compromise between models' performances and diversity was made and the set of selected hydrological models were MORDOR-SD, GRSD, SMASH, SIM2 et J2000 for high flows, MORDOR-SD, GRSD, SMASH et SIM2 for mean flows, and MORDOR-SD, GRSD, SMASH and J2000 for low flows.

It is worth pointing out that the discrepancies observed between reference data and simulated data are not solely attributable to the models. The analysis relied on reference datasets. Despite the care taken during the selection of reference points and the filtering of outlier data, reference datasets may still contain errors or residual influences. In addition, forcings in mountainous areas (which are inherently heterogeneous) may lack precision or, more generally, may fail to capture convective precipitation.

Before any use of hydrological projections, it should be recalled that strong model performance under present-day conditions does not necessarily imply reliability under climate change. This limitation is particularly pronounced for empirically based or

conceptual models, which rely on empirical relationships and parameterizations calibrated to recent climate conditions. Such models are often finely tuned to reproduce observed behavior as closely as possible. This is, for example, the case for conceptual hydrological models in which snowmelt is represented using temperature-index approaches (e.g., degree-day methods) and evapotranspiration losses are estimated from potential evapotranspiration formulations. As a result, these carefully calibrated models often exhibit high performance and may outperform physically based models that represent processes more explicitly. However, the ability to reproduce past observations does not guarantee that a model will adequately represent processes under altered hydroclimatic conditions. Model evaluation is therefore a critical issue in model selection. While it must assess the capacity to reproduce observations, it should also examine—when possible—the temporal transferability of models. This second aspect of evaluation is difficult, and often impossible, but remains essential. A particularly critical issue concerns evapotranspiration losses. The feedbacks of increasing atmospheric $CO_2$ on plant phenology and, consequently, on evapotranspiration are typically not represented in hydrological models. These feedbacks may lead to substantially different future water balances. Physically based models that allow exploration of such interactions therefore deserve consideration, even if their performance is lower than that of highly calibrated empirical models. Ongoing climate and hydrological changes may provide opportunities to assess model behavior under modified conditions, although targeted and enhanced observations will likely be required to support such evaluations.

In an ideal world, the research community should examine the divergences between models driven by the same climate projections and investigate their origins (e.g. by running additional experiments). For example, in Explore2, the hydrological model ORCHIDEE shows important discrepancies with the other hydrological models for many catchments. In continuation of Explore2, developers of the model ORCHIDEE have clearly identified what parts of the land surface processes could be better represented (groundwater module for drainage, snow representation in mountainous catchments, Huand et al., 2024). On the other hand, developers of conceptual models have also many perspectives to improve the transferability of their modelling framework for future climates, in particular regarding the evolution of the vegetation (land use, vegetation type, interactions between vegetation and the carbon cycle).

Héraut, Louis; Vidal, Jean-Philippe; Évin, Guillaume; Sauquet, Éric,"Notice de lecture des fiches de résultats des modèles hydrologiques de surface", Recherche Data Gouv, 2024. INRAE. https://doi.org/10.57745/6YNIUF

Huang, Peng, Agnès Ducharne, Lucia Rinchiuso, et al. « Multi-Objective Calibration and Evaluation of the ORCHIDEE Land Surface Model over France at High Resolution ». *Hydrology and Earth System Sciences* 28, nᵒ 19 (2024): 4455‑76. https://doi.org/10.5194/hess-28-4455-2024.

Sauquet, Éric, Louis Héraut, Jérémie Bonneau, Alix Reverdy, Laurent Strohmenger, and Jean-Philippe Vidal. *Diagnostic des modèles hydrologiques : Des données aux résultats*. Recherche Data Gouv, 2025. INRAE. https://doi.org/10.57745/S6PQXD

Sauquet, E, and L. Héraut. « Notice de lecture des fiches « diagnostic » des modèles hydrologiques ». Recherche Data Gouv, 2024, INRAE. https://doi.org/10.57745/MDHS0D.

**Reviewing #3**

**3.1. The manuscript titled Uncertainty sources in a large ensemble of hydrological projections: Regional Climate Models and Internal Variability matter by Evin et al. is well written and presents a comprehensive and carefully structured discussion, along with clearly articulated limitations and conclusions. I particularly appreciated the depth and clarity of the discussion section. The proposed methodology provides a robust framework to quantify uncertainties arising from multiple sources, including GCMs, RCMs, HMs, and internal variability, across different hydrological regimes, which I found both insightful and highly relevant.**

The discussion provides a comprehensive interpretation of the results and highlights key findings regarding the relative contributions of individual models to overall uncertainty. In particular, the finding such as that one or a small number of models can contribute disproportionately to the total uncertainty is both interesting and well-supported. A deeper investigation into the underlying reasons for inter-model differences or the identification of hydrological conditions under which certain models may perform better could form the basis of a separate study. Nevertheless, by focusing solely on model output (data alone), the present work opens up a wide range of future research directions.

Overall, the manuscript is scientifically sound, clearly presented, and well-motivated, and I wholeheartedly recommend it for publication.

We sincerely thank the reviewer for this very encouraging and positive feedback.